# Enhancing Adversarial Robustness with Conformal Prediction: A Framework for Guaranteed Model Reliability

Jie Bao [1 2]   Chuangyin Dang [3]   Rui Luo [† 2 3]   Hanwei Zhang [4]   Zhixin Zhou [5]

## Abstract

As deep learning models are increasingly deployed in high-risk applications, robust defenses against adversarial attacks and reliable performance guarantees become paramount. Moreover, accuracy alone does not provide sufficient assurance or reliable uncertainty estimates for these models. This study advances adversarial training by leveraging principles from Conformal Prediction. Specifically, we develop an adversarial attack method, termed OPSA (OPtimal Size Attack), designed to reduce the efficiency of conformal prediction at any significance level by maximizing model uncertainty without requiring coverage guarantees. Correspondingly, we introduce OPSA-AT (Adversarial Training), a defense strategy that integrates OPSA within a novel conformal training paradigm. Experimental evaluations demonstrate that our OPSA attack method induces greater uncertainty compared to baseline approaches for various defenses. Conversely, our OPSA-AT defensive model significantly enhances robustness not only against OPSA but also other adversarial attacks, and maintains reliable prediction. Our findings highlight the effectiveness of this integrated approach for developing trustworthy and resilient deep learning models for safety-critical domains. Our code is available at https://github.com/bjbbbb/Enhancing-Adversarial-Robustness-with-Conformal-Prediction.

## 1. Introduction

Recent advancements in Deep Neural Networks (DNNs) have demonstrated remarkable efficacy across diverse domains (He et al., 2016b; Gupta & Verma, 2023b;a). Despite their success, a critical challenge remains in the precise quantification of predictive uncertainty during real-world deployment. Addressing this challenge, Conformal Prediction (CP) (Vovk et al., 2005) has emerged as a promising paradigm, distinguished by its distribution-free properties and robust uncertainty quantification capabilities. This data-driven methodology excels in both regression (Luo & Zhou, 2025a) and classification (Luo & Zhou, 2024) tasks by providing statistically rigorous confidence intervals and well-calibrated probability estimates. Consequently, CP effectively characterizes inherent data uncertainty while maintaining controlled classification error rates. Moreover, through the implementation of defensive mechanisms, CP substantially enhances model robustness against adversarial perturbations (Li et al., 2024), highlighting its significant potential for deployment in safety-critical applications such as autonomous driving (Doula et al., 2024) and medical diagnosis (Luo et al., 2024).

In the context of CP, current research efforts in adversarial robustness model primarily focus on adjusting nonconsistent scoring functions to mitigate the impact of adversarial attacks on test or calibration data (Gendler et al., 2021; Einbinder et al., 2022; Ghosh et al., 2023; Yan et al., 2024; Cauchois et al., 2024). However, these approaches present significant limitations. Firstly, ensuring comprehensive coverage often requires substantially increasing the size of the prediction set, which inevitably reduces the practical value and accuracy of the prediction results. Secondly, these methods are typically optimized for specific types of adversarial attacks (Croce et al., 2021), rendering their robustness vulnerable when faced with unknown or novel attack vectors. Additionally, existing techniques exhibit notable deficiencies in computational efficiency and generalization ability, significantly hindering their widespread adoption and practical application.

Some CP methods enhance adversarial robustness even further by integrating adversarial training (AT) algorithms. In these algorithms, adversarial training is conceptualized as a

---

[1]Huaiyin Institute of Technology, Huai'an, China [2]Chengdu Research Institute, City University of Hong Kong, Chengdu, China [3]City University of Hong Kong, Hong Kong, China [4]Saarland University, Saarbrücken, Germany [5]Alpha Benito Research, Los Angeles, USA. Correspondence to: Rui Luo <ruiluo@cityu.edu.hk>.

two-player zero-sum game, where a defender and an adversary strive to minimize and maximize classification errors, respectively (Nouiehed et al., 2019). However, the inherent discontinuity of classification errors poses challenges for first-order optimization algorithms, making the practical implementation of this zero-sum framework difficult. Recently, (Robey et al., 2024) redefined adversarial training as a non-zero-sum game by pursuing distinct objectives, thereby advancing the training process.

Despite these advancements, the application of Conformal Prediction (CP) within adversarial settings presents unique challenges (Gendler et al., 2021). Specifically, within the CP framework, the primary objective is to minimize the size of the prediction set while ensuring statistical coverage. Achieving this balance is crucial, as overly large prediction sets can reduce the practical utility and interpretability of the model's outputs. However, there is a scarcity of research that directly addresses the optimization of prediction set size within CP, especially in adversarial contexts. Most existing studies focus on maintaining coverage under adversarial perturbations, often at the expense of significantly enlarging the prediction sets (Ghosh et al., 2023). Notably, (Stutz et al., 2021) introduced conformal training methods aimed at more effectively minimizing the prediction set size without compromising coverage guarantees. This approach represents a promising direction for future work, as it seeks to enhance the practical applicability of CP by ensuring that the generated prediction sets are both accurate and manageable in size.

To address the aforementioned challenges, we propose an adversarial training algorithm within the Conformal Prediction framework, aimed at minimizing the prediction set size while ensuring coverage. Specifically, our contributions are as follows:

- We design a differentiable and smooth attack function based on negative gradients. This attack method maximizes the size of the prediction set by introducing imperceptible perturbations.

- We develop a CP defense model leveraging adversarial attacks with theoretical guarantees. By partitioning the training set into two subsets—one dedicated to ensuring coverage and the other to minimize the prediction set size—we address this as a bi-objective optimization problem.

- Through experiments CIFAR-10, CIFAR-100 and mini-ImageNet datasets, we demonstrate that our attack method generates the largest prediction set sizes compared to existing attack techniques, thereby increasing model uncertainty. Additionally, our adversarially trained model effectively minimizes model uncertainty and enhances robustness relative to other methods.

## 2. Related Work

Conformal prediction (CP) (Vovk et al., 2005) is a methodology designed to generate prediction regions for variables of interest, thereby enabling the estimation of model uncertainty by substituting point predictions with prediction regions. This methodology has been widely applied in both classification (Luo & Zhou, 2024; Luo & Colombo, 2024; Luo & Zhou, 2025b) and regression tasks (Luo & Zhou, 2025e;f). Furthermore, CP can be adapted to diverse real-world scenarios, including segmentation (Luo & Zhou, 2025c), time-series forecasting (Su et al., 2024), and graph-based applications (Luo et al., 2023; Tang et al., 2025; Luo & Zhou, 2025d; Wang et al., 2025; Luo & Colombo, 2025; Zhang et al., 2025).

**Adversarial Attack against Conformal Prediction** The emergence of adversarial phenomena (Goodfellow et al., 2014; Zhang et al., 2022) has raised significant security concerns in machine learning. Uncertainty estimation plays a vital role in ensuring the robustness of deep learning models. Conformal Prediction (CP) (Vovk et al., 2005), offers distribution-free coverage guarantees but encounters difficulties when subjected to data poisoning and adversarial attacks. Studies such as Liu et al. (2024) demonstrate that standard adversarial attack techniques, including PGD (Madry et al., 2017), can effectively compromise the robustness of conformal prediction. Kumar et al. (2024) provide a survey and comparative analysis of robust conformal prediction methods, highlighting their strengths and limitations.

**Adversarially Robust Conformal Prediction** To mitigate the adversarial impact on CP, a series of studies have attempted to address this issue without involving training. Adversarially Robust Conformal Prediction (ARCP) (Gendler et al., 2021) combines conformal prediction with randomized smoothing to ensure finite-sample coverage guarantees under $L_2$-norm-bounded adversarial noise. It leverages Gaussian noise to bound the Lipschitz constant of the nonconformity score, addressing unknown adversarial perturbations without training. Probabilistically Robust Conformal Prediction (PRCP) (Ghosh et al., 2023) adapts to perturbations using a quantile-of-quantile design, determining thresholds for both data samples and perturbations. It uses adversarial attacks to calculate empirical robust quantiles, independent of model training. Yan et al. (2024) propose Post-Training Transformation (PTT) and Robust Conformal Training (RCT) to improve the efficiency of robust conformal prediction. They modify RSCP into RSCP+ for certified guarantees and embed it into the training process. Zargarbashi et al. (2024) derive robust prediction sets by bounding worst-case changes in conformity scores for adversarial evasion and poisoning attacks. They use CDF-based bounds to compute conservative prediction sets and thresholds for

these scenarios. Jeary et al. (2024) introduce Verifiable Robust Conformal Prediction (VRCP), leveraging neural network verification to maintain coverage guarantees under adversarial attacks. VRCP supports arbitrary norm-bounded perturbations and extends to regression tasks. These methods partially mitigate the adversarial impact; however, they either compromise the compactness of the prediction set or fail to maintain robustness against different types of attacks and perturbation sizes.

**Adversarial Training for Conformal Prediction** To further enhance adversarial robustness, a natural approach is to incorporate adversarial training into CP. Liu et al. (2024) propose Uncertainty-Reducing Adversarial Training (AT-UR) to improve CP efficiency and adversarial robustness by minimizing predictive entropy and using a weighted loss based on True Class Probability Ranking. They integrate AT-UR with AT (Madry et al., 2017), FAT (Zhang et al., 2020), and TRADES (Zhang et al., 2019), using adversarial examples generated by PGD (Kurakin et al., 2016). Luo et al. (2024) propose training specialized defensive models tailored to specific attack types and utilizing maximum and minimum classifiers to effectively combine these defenses. However, existing adversarial training methods for CP are limited to specific attacks due to the challenge of solving the max-min optimization in this context. To address this, we reformulate it as a bi-level optimization, making our method attack-agnostic.

## 3. Method

Consider a classifier network $f : \mathcal{X} \to \mathbb{R}^K$ that maps an input image $\mathbf{x} \in \mathcal{X}$ to a logit vector $f(\mathbf{x}) \in \mathbb{R}^K$. Here, $\mathcal{X} = [0, 1]^d$ represents the image space, where pixel values are normalized and $d$ denotes the image dimension. The integer $K$ signifies the total number of classes, and $y \in [K] := \{1, \ldots, K\}$ denotes the ground truth label for the input $\mathbf{x}$.

### 3.1. Conformal Prediction

For a given classifier $f$ and an input $\mathbf{x}$, a prediction set can be formed by selecting all classes $k$ whose corresponding logit $f_k(\mathbf{x})$ exceeds a certain threshold $\tau$. This prediction set is defined as:

$$\Gamma(\mathbf{x}; f, \tau) = \{k \in [K] : f_k(\mathbf{x}) \geq \tau\}. \quad (1)$$

The THR method for conformal prediction (Sadinle et al., 2019) determines this threshold $\tau$ as the $(1 - \alpha)$-quantile, denoted $q_{1-\alpha}$, of conformity scores computed on a separate calibration set. To be more precise, if $\{(\mathbf{x}_i, y_i)\}$ for $i \in \mathcal{I}_{\text{cal}}$

is the calibration set, then

$$q_{1-\alpha} = \lceil (1 + |\mathcal{I}_{\text{cal}}|)(1 - \alpha) \rceil \text{-th largest value in}$$
$$\{f_{y_i}(\mathbf{x}_i) : i \in \mathcal{I}_{\text{cal}}\}.$$

The resulting prediction set, $\Gamma(\mathbf{x}; f, q_{1-\alpha})$, then guarantees marginal coverage:

$$\mathbb{P}(y \in \Gamma(\mathbf{x}; f, q_{1-\alpha})) \geq 1 - \alpha. \quad (2)$$

This means that, under the assumption of exchangeability between calibration and test data, the prediction set includes the true label $y$ with a probability of at least $1 - \alpha$.

*Remark* 1 (Choice of Non-conformity Score). In the preceding discussion and our proposed framework, we primarily adopt the specific non-conformity score $s(\mathbf{x}, k) = 1 - f_k(\mathbf{x})$ as utilized in the Threshold Response (THR) method (Sadinle et al., 2019). It is important to note that our methodology is not inherently limited to this particular score. A variety of other non-conformity score functions exist, such as those proposed in (Romano et al., 2020; Luo & Zhou, 2024; 2025b). The core principles of our attack and defense strategies can be readily extended to accommodate alternative score functions $s_f(\mathbf{x}, y)$, provided that these functions are differentiable (or at least sub-differentiable) with respect to the model $f$'s outputs (or more precisely, with respect to the values $f_k(\mathbf{x})$ upon which $s_f$ depends). Our choice of the THR score function is primarily motivated by its potential to yield efficient prediction sets under standard conditions. Furthermore, the resulting score for the true class, $s(\mathbf{x}, y) = 1 - f_y(\mathbf{x})$, simplifies the notation and derivation within our framework.

### 3.2. Adversarial Training as a Min-Max Problem.

Adversarial training is often conceptualized as a min-max optimization problem, aiming to find model parameters $\theta$ that minimize the loss on adversarially perturbed inputs. This saddle-point problem can be formulated as (Madry et al., 2017):

$$\min_{\theta} \mathbb{E}_{(\mathbf{x},y) \sim \mathcal{D}} \left[ \max_{\|\epsilon\|_p \leq r} \ell(\theta, \mathbf{x} + \epsilon, y) \right]. \quad (3)$$

Here, $(\mathbf{x}, y)$ represents a data point sampled from the distribution $\mathcal{D}$, consisting of an input $\mathbf{x}$ and its true label $y$. The term $\epsilon$ denotes an adversarial perturbation, constrained within a specific norm-ball (e.g., $\|\epsilon\|_p \leq r$, where $p$ could be $\infty$ or 2, and $r$ is the perturbation budget). The function $\ell(\theta, \mathbf{x} + \epsilon, y)$ is the loss incurred by the model with parameters $\theta$ on the perturbed input $\mathbf{x} + \epsilon$ with respect to the true label $y$. The choice of the loss function $\ell$ can vary depending on the specific task. In this paper, we will adopt a loss function for conformal prediction task.

### 3.3. Adversarial Attack against Conformal Prediction

An adversarial perturbation $\epsilon$ against conformal prediction aims to disrupt the prediction set $\Gamma(f(\mathbf{x}+\epsilon), \tau)$ of classifier $f$ while remaining imperceptible. The adversarial attack seeks to achieve the following objectives:

- **Maximize uncertainty:** Increase the expected size of the conformal prediction set $\mathbb{E}\left[|\Gamma\left(f(\mathbf{x}+\epsilon), \tau\right)|\right]$ to reduce the informativeness of predictions at any threshold $\tau$. $\tau$ is unknown since the attacker do not know the significance level of the defender.

- **Maintain imperceptibility:** Ensure the perturbation satisfies the constraint $\|\epsilon\|_p \leq r$ for a predefined budget $r$ to remain undetectable.

We propose the following objective for fixed classifier network but it does not involve a significance level $\alpha$. We first define soft set size

$$M_T(\mathbf{x}; f, \tau) = \sum_{k \in [K]} \sigma\left(\frac{f_k(\mathbf{x}) - \tau}{T}\right), \qquad (4)$$

where $\sigma(x) = 1/(1 + e^{-z})$ is the sigmoid function and $T > 0$ is a temperature hyperparameter. It is worth noting that for fixed $(\mathbf{x}; f, \tau)$ such that $f_k \neq \tau$ for every $k \in [K]$,

$$\lim_{T \downarrow 0} \sigma\left(\frac{f_k(\mathbf{x}) - \tau}{T}\right) = \mathbf{1}\{k \in \Gamma(\mathbf{x}; f, \tau)\}, \qquad (5)$$

and

$$\lim_{T \downarrow 0} M_T(x; f, \tau) = |\Gamma(\mathbf{x}; f, \tau)|. \qquad (6)$$

The sigmoid function and $M_T$ represent the soft indicator function and set size function respectively. This is a technique also employed in conformal training methodologies (Stutz et al., 2021). For a given input sample $(\mathbf{x}, y)$, we propose that the attacker's objective is to find an adversarial perturbation $\epsilon^*(\mathbf{x}, y)$ that maximizes this soft set size, using the perturbed true class score $f_y(\mathbf{x} + \epsilon)$ as the internal reference threshold within $M_T$. The optimization problem is thus formulated as:

$$\epsilon^*(\mathbf{x}, y) = \underset{\substack{\epsilon:\|\epsilon\|_p \leq r, \\ \mathbf{x}+\epsilon \in [0,1]^d}}{\arg\max} M_T(\mathbf{x} + \epsilon; f, f_y(\mathbf{x} + \epsilon)). \quad (7)$$

Here, the maximization is performed for each specific sample $(\mathbf{x}, y)$ to find its corresponding optimal perturbation.

The construction of this objective function shares conceptual similarities with prior work, such as (Robey et al., 2024), particularly in its implicit use of logit differences of the form $f_k(\mathbf{x} + \epsilon) - f_y(\mathbf{x} + \epsilon)$. Our approach distinctively

---

**Algorithm 1** Optimal Size Attack (OPSA)

**Require:** A single labeled data: $(\mathbf{x}, y)$;
     Perturbation budget: $r$; Temperature: $T_1$;
     Maximum iteration number: $J$;
     Targeted classifier: $f$;
     Learning rate: $\eta$.
**Ensure:** Adversarial perturbation: $\epsilon^*$.
 1: **function** OPSA($\mathcal{D}_{\text{train}}, r, f, T_1, J, \eta$)
 2:  ▷ Initialize Perturbation:
 3:  $\epsilon \longleftarrow \text{Unif}(B_r(0))$.
 4:  **while** $j \leq J$ or $\epsilon$ has not converged **do**
 5:   ▷ Update Perturbation:
 6:   $\epsilon \longleftarrow \epsilon + \eta \nabla_\epsilon M_{T_1}(\mathbf{x} + \epsilon; f, f_y(\mathbf{x} + \epsilon))$
 7:   $\epsilon \longleftarrow \Pi_{B_r(0) \cap ([0,1]^d - x)}$
 8:   $j \longleftarrow j + 1$
 9:  **end while**
 10:  **Return** $\epsilon^*$
 11: **end function**

---

applies a scaled sigmoid function to these effective differences (achieved by setting the internal threshold of $M_T$ to $f_y(\mathbf{x} + \epsilon)$) to specifically target the maximization of the (soft) prediction set size. A key advantage of this formulation is that the attacker does not require knowledge of the defender's chosen significance level $\alpha$ (and consequently, the operational threshold $\tau$) to craft the perturbation $\epsilon^*$.

Algorithm 1 details the iterative process for finding the optimal perturbation $\epsilon^*$ for a given input sample. Although presented conceptually as gradient ascent for simplicity, our actual implementation leverages the Adam optimizer for more effective and stable optimization, leading to higher-quality adversarial perturbations. A critical component of each iteration is the projection of the updated perturbation $\epsilon$ back onto the feasible region. This step guarantees that the perturbation satisfies both the norm constraint, $\|\epsilon\|_p \leq r$ (confining $\epsilon$ to the ball $B_r(0)$), and the image validity constraint, ensuring $\mathbf{x} + \epsilon \in [0, 1]^d$. The set of perturbations satisfying the image validity constraint can be expressed as $[0, 1]^d - \mathbf{x} = \{\epsilon \in \mathbb{R}^d : \mathbf{x} + \epsilon \in [0, 1]^d\}$.

### 3.4. Adversarial Robust Conformal Training

In the context of the min-max formulation presented in (3), adversarial training (AT) endeavors to identify a classifier $f(\mathbf{x}; \theta)$, parameterized by $\theta$, that minimizes the loss incurred from potential adversarial perturbations $\epsilon$ applied to the input images. Our work posits that for achieving robust conformal prediction, it is particularly crucial to train the classifier against the specific perturbation $\epsilon^*$ generated by our OPSA method (detailed in Algorithm 1). The training of this classifier, $f(\cdot; \theta)$, is subsequently guided by the two loss components detailed below.

**Classification Loss.** The classification loss, denoted as $\mathcal{L}_{\text{class}}$, is designed to encourage the inclusion of the true class label $y$ within the soft prediction set (implicitly defined by $\tau$ and $T_2$), while simultaneously penalizing the inclusion of incorrect classes. For a given input $\mathbf{x}$, its true label $y$, classifier $f$, threshold $\tau$, and temperature $T_2$, this loss is defined as:

$$\mathcal{L}_{\text{class}}\left(\mathbf{x}, y; f, \tau, T_2\right) = \sigma\left(\frac{f_y(\mathbf{x}) - \tau}{T_2}\right) - \sum_{k \neq y} \sigma\left(\frac{f_k(\mathbf{x}) - \tau}{T_2}\right)$$

To understand the behavior of this loss, consider the limit as $T_2 \downarrow 0$, where the sigmoid function $\sigma(\cdot)$ approximates an indicator function $I(\cdot \geq 0)$. In this scenario, the first term, $\sigma\left(\frac{f_y(\mathbf{x}) - \tau}{T_2}\right)$, approaches 1 if $f_y(\mathbf{x}) \geq \tau$ (i.e., the true label is correctly included in the hard prediction set), and 0 otherwise. The second term, $\sum_{k \neq y} \sigma\left(\frac{f_k(\mathbf{x}) - \tau}{T_2}\right)$, approximates the count of incorrect classes $k \neq y$ for which $f_k(\mathbf{x}) \geq \tau$ (i.e., the number of incorrect labels erroneously included in the hard prediction set). Thus, $\mathcal{L}_{\text{class}}$ effectively approximates the difference between an indicator for the true label's inclusion and the count of incorrectly included labels. Maximizing this loss (or minimizing its negative) encourages both the inclusion of $y$ and the exclusion of $k \neq y$ from the prediction set defined by $\tau$.

**Size Loss.** The size loss, denoted as $\mathcal{L}_{\text{size}}$, aims to minimize the average size of the confidence sets, thereby reducing inefficiency. The soft prediction set size is defined in (4). We will use the same notation $M_T$ and its definition.

**Total Loss Function.** To balance the contributions of classification accuracy and the efficiency of the confidence sets, the total loss function integrates the classification loss and the size loss using a weighting factor $\lambda$:

$$\begin{aligned} \mathcal{L}_{\text{total}}(\mathbf{x}, y; \theta, \tau, T_2) &= \mathcal{L}_{\text{class}}(\mathbf{x}, y; f(\cdot; \theta), \tau, T_2) \\ &\quad + \lambda M_{T_2}(\mathbf{x}; f(\cdot; \theta), \tau), \end{aligned} \quad (8)$$

where $\theta$ are the parameters of $f$.

**Adversarial Training Procedure.** The algorithm trains the model on noisy images subjected to the OPSA attack. We assume the defender possesses noise-free images with their corresponding labels. The perturbation $\epsilon^*$, as the output of Algorithm 1, is added to the images. Differing from some conformal training approaches such as (Stutz et al., 2021), the threshold $\tau$ in our method is obtained by computing the exact quantile, rather than being determined via a sigmoid-approximated mechanism. The objective is then to find $\theta^*$ that minimizes the loss function defined in (8):

$$\theta^* := \arg\min_{\theta} \sum_{i \in \mathcal{I}_{\text{train}}} \mathcal{L}_{\text{total}}(\mathbf{x}_i + \epsilon_i^*, y_i; \theta, \tau, T_2).$$

---

**Algorithm 2** OPSA Adversarial Training (OPSA-AT)

**Require:** A set of labeled data: $\mathcal{D}_{\text{train}}$;
      Pre-trained classifier parameter: $f_{\text{init}}$;
      Pre-determined Coverage Probability: $1 - \alpha$;
      Maximum number of epoch: $J'$;
      Temperature: $T_2$; Loss weight: $\lambda$;
      Parameters for adversarial attack: $r, T_1, J$.
**Ensure:** Robust conformal prediction model $f^*$.
1: **function** OPSA-AT($\mathcal{I}_{\text{train}}, f, T_2, K', \lambda$)
2:    ▷ Initialize classifier parameter:
3:    $\theta \leftarrow \theta_{\text{init}}$
4:    **while** $j \leq J'$ or $\theta$ has not converged **do**
5:      **for** mini-Batch $\mathcal{B} \subset \mathcal{D}_{\text{train}}$ **do**
6:        Randomly split $\mathcal{B}$ into $\mathcal{B}_{\text{train}}$ and $\mathcal{B}_{\text{cal}}$.
7:        ▷ Train attacker's perturbation:
8:        **for** $i \in \mathcal{B}_{\text{train}}$ **do**
9:          $\epsilon_i^* \longleftarrow \text{OPSA}((\mathbf{x}_i, y_i), r, f^{\text{AT}}(\cdot, \theta), T_1, J, \eta)$.
10:       **end for**
11:       ▷ Find threshold:
12:       $\tau \longleftarrow$ as the $\lceil(1 + |\mathcal{B}_{\text{cal}}|)(1 - \alpha)\rceil$ largest $f_{y_i}(\mathbf{x}_i; \theta)$ for $(\mathbf{x}_i, y_i) \in \mathcal{B}_{\text{cal}}$.
13:       ▷ Update classifier parameter:
14:       $\mathcal{I} \longleftarrow$ set of indices of $\mathcal{B}_{\text{train}}$
15:       $\theta \leftarrow \theta - \eta \nabla_\theta \sum_{i \in \mathcal{I}} \mathcal{L}_{\text{total}}(\mathbf{x}_i + \epsilon_i^*, y_i; \theta, \tau, T_2)$
16:       $j \longleftarrow j + 1$
17:      **end for**
18:    **end while**
19:    $\theta^* \longleftarrow \theta$
20:    **Return** $f^{\text{AT}}(\cdot, \theta^*)$
21: **end function**

---

The optimization of $\theta$ (model training) and the determination of the threshold $\tau$ (calibration) are performed iteratively, as the optimal $\tau$ depends on the current model $f(\cdot; \theta)$, and the update to $\theta$ depends on $\tau$.

Algorithm 2 outlines the steps of our adversarial robust conformal training. In addition to the optimization procedure for $\theta$, the algorithm first specifies the mini-batch determination during the training process. For each batch, the data (now adversarial) is further split into training and calibration subsets. The adversarial perturbation $\epsilon^*$ is generated based on the data designated for the training subset of the batch, and the threshold $\tau$ is subsequently obtained from the calibration subset of the same batch.

### 3.5. Conformal Prediction with the Robust Classifier

Having obtained a robust classifier, $f_{\text{AT}}(\cdot; \theta^*)$, through the adversarial training procedure (OPSA-AT), we now integrate it with the conformal prediction methodology. The application of conformal prediction largely mirrors the procedure detailed in Section 3.1, with the crucial substitution

of the original classifier $f(\mathbf{x})$ with our adversarially trained classifier $f_{\text{AT}}(\mathbf{x}; \theta^*)$. The following key aspects underpin this integration.

**Data Splitting.** A fundamental prerequisite for valid conformal guarantees is that the dataset used for training the model parameters $\theta^*$ (i.e., the adversarial training set) must be strictly disjoint from the samples utilized for the CP calibration phase. This separation is crucial for the integrity of the CP guarantees.

**Exchangeability for Calibration and Test Data.** The validity of CP hinges on the core assumption of exchangeability between the calibration and test samples. It is important to emphasize that this exchangeability assumption does not impose specific requirements on whether these samples are adversarially perturbed or not. For instance, the CP guarantees hold if all calibration and test samples are clean (unperturbed), if all are perturbed (e.g., by the OPSA attack detailed in Algorithm 1), or if a random subset of them is perturbed, as long as the perturbation mechanism (or lack thereof) is applied consistently or randomly in a way that preserves exchangeability between the two sets. The validity of CP relies solely on this exchangeability property of the (potentially transformed) data points.

**Experimental Focus.** In the experimental evaluations presented in this paper, we specifically investigate a challenging scenario. While the adversarial training for $\theta^*$ might be based on or include clean samples, both the calibration and test datasets are composed of samples adversarially perturbed by our OPSA method. This setup stringently tests the robustness of the conformalized predictions.

# 4. Experiments

**Settings.** In this section, we evaluate the performance of OPSA and OPSA-AT on the CIFAR-10 (Krizhevsky et al., 2009) and CIFAR-100 datasets using ResNet34 (He et al., 2016a) as the training framework. Additionally, we conduct evaluations on the mini-ImageNet (Deng et al., 2009; Vinyals et al., 2016) dataset using ResNet50. For attack, we compare OPSA to methods such as FGSM (Goodfellow et al., 2014), PGD[10] (Madry et al., 2017), PGD[40], BETA[10] (Robey et al., 2024), Square (1000 queries) (Andriushchenko et al., 2020), Auto[100] attack (Croce & Hein, 2020), and APGD[100] (Croce & Hein, 2020), where the superscript indicates the attack iteration. For defense, we select adversarial training based on FGSM, PGD, as well as TRADES (Zhang et al., 2019), MART (Wang et al., 2019), and BETA-AT for comparison, with each model undergoing 10 iterations. For parameter settings, we adhere to the standard perturbation budget of $\epsilon = 8/255$ (0.03) and $\ell_\infty$ norm, applying a step size of $2/255$ attacks during training and

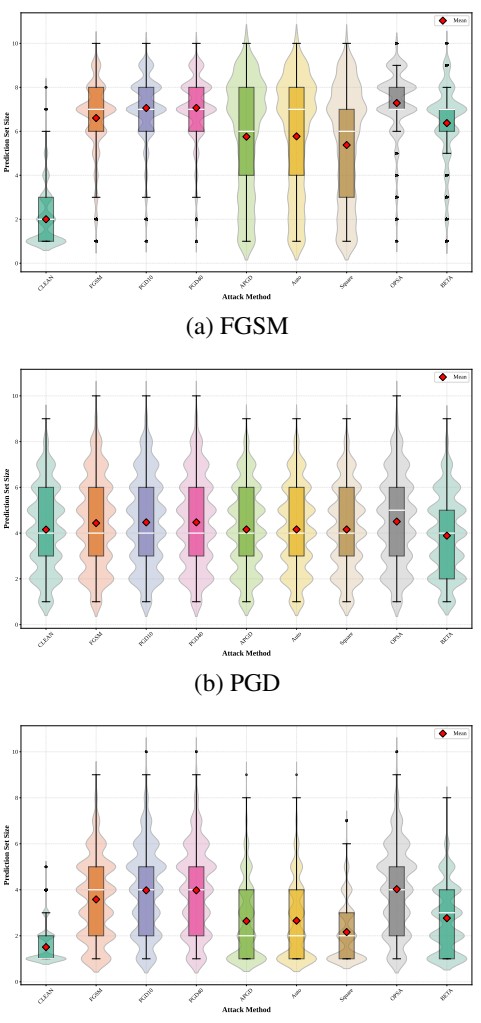

(a) FGSM

(b) PGD

(c) TRADES

*Figure 1.* Box-violin plots of CIFAR-10 results under FGSM, PGD, and TRADES defense models

testing. The trade-off parameter for TRADES and MART is set to 5, in alignment with their original implementations. To initialize our models, We conduct pre-training on clean data for 5 epochs on the CIFAR$-10$ and CIFAR$-100$ datasets, while for the mini-ImageNet dataset, we perform 10 epochs of pre-training on clean data. Subsequently, on the CIFAR$-10$ and CIFAR$-100$ datasets, we split the test set into $\mathcal{I}_{\text{cal}}$ and $\mathcal{I}_{\text{test}}$ in a ratio of 20% to 80%. Given that the training and test sets of mini-ImageNet are not interchangeable, we divide its training set into $\mathcal{I}_{\text{train}}$, $\mathcal{I}_{\text{cal}}$, and $\mathcal{I}_{\text{test}}$ in proportions of 50%, 25%, and 25%, respectively. Following (Stutz et al., 2021), we set both $T1$ and $T2$ to 1 on the CIFAR$-10$, CIFAR$-100$, and Mini-ImageNet datasets to approximate the THR method. We evaluate model robustness by launching attacks on both validation and test sets, setting the alpha parameter to 10%.

| Attacks | Indicator | Training Algorithm | | | | | | |
|---|---|---|---|---|---|---|---|---|
| | | **FGSM** | **PGD**[10] | **TRADES**[10] | **MART**[10] | **BETA-AT**[10] | **OPSA-ST**[10] | **OPSA-AT**[10] |
| Clean | Coverage (%) | $90.50 \pm 0.33$ | $89.40 \pm 0.34$ | $89.72 \pm 0.35$ | $89.44 \pm 0.34$ | $88.24 \pm 0.36$ | $89.49 \pm 0.31$ | $89.25 \pm 0.34$ |
| | Size | $2.00 \pm 0.01$ | $4.15 \pm 0.02$ | $1.51 \pm 0.01$ | $1.29 \pm 0.01$ | $7.06 \pm 0.03$ | $1.30 \pm 0.01$ | $\underline{1.24 \pm 0.01}$ |
| | SSCV | $0.04 \pm 0.01$ | $\underline{0.02 \pm 0.01}$ | $0.04 \pm 0.01$ | $0.10 \pm 0.01$ | $0.11 \pm 0.03$ | $0.08 \pm 0.03$ | $\mathbf{0.08 \pm 0.01}$ |
| FGSM | Coverage (%) | $87.99 \pm 0.37$ | $88.88 \pm 0.38$ | $89.05 \pm 0.34$ | $89.08 \pm 0.35$ | $90.05 \pm 0.66$ | $89.38 \pm 0.35$ | $89.48 \pm 0.35$ |
| | Size | $6.61 \pm 0.02$ | $4.43 \pm 0.02$ | $3.59 \pm 0.02$ | $4.61 \pm 0.02$ | $3.20 \pm 0.02$ | $2.56 \pm 0.02$ | $\underline{2.50 \pm 0.02}$ |
| | SSCV | $0.03 \pm 0.01$ | $0.03 \pm 0.01$ | $\underline{\mathbf{0.02 \pm 0.01}}$ | $0.03 \pm 0.01$ | $0.08 \pm 0.07$ | $0.02 \pm 0.01$ | $0.03 \pm 0.01$ |
| PGD[10] | Coverage (%) | $88.24 \pm 0.36$ | $88.84 \pm 0.36$ | $89.08 \pm 0.35$ | $89.36 \pm 0.34$ | $88.90 \pm 0.34$ | $89.19 \pm 0.34$ | $89.29 \pm 0.35$ |
| | Size | $7.06 \pm 0.02$ | $4.46 \pm 0.02$ | $3.96 \pm 0.02$ | $5.19 \pm 0.02$ | $6.63 \pm 0.02$ | $\mathbf{3.46 \pm 0.03}$ | $\underline{\mathbf{3.24 \pm 0.03}}$ |
| | SSCV | $\underline{0.02 \pm 0.01}$ | $0.03 \pm 0.01$ | $0.02 \pm 0.01$ | $0.05 \pm 0.01$ | $0.06 \pm 0.02$ | $0.07 \pm 0.01$ | $0.07 \pm 0.01$ |
| PGD[40] | Coverage (%) | $88.33 \pm 0.37$ | $88.84 \pm 0.34$ | $89.35 \pm 0.35$ | $89.04 \pm 0.35$ | $88.92 \pm 0.35$ | $89.20 \pm 0.35$ | $89.30 \pm 0.03$ |
| | Size | $7.07 \pm 0.02$ | $4.47 \pm 0.02$ | $3.97 \pm 0.02$ | $5.19 \pm 0.02$ | $6.84 \pm 0.03$ | $3.46 \pm 0.03$ | $\underline{3.24 \pm 0.03}$ |
| | SSCV | $0.02 \pm 0.03$ | $0.03 \pm 0.01$ | $\underline{0.02 \pm 0.01}$ | $0.05 \pm 0.02$ | $0.08 \pm 0.02$ | $0.07 \pm 0.01$ | $0.07 \pm 0.01$ |
| BETA[10] | Coverage (%) | $88.05 \pm 0.37$ | $89.21 \pm 0.35$ | $89.06 \pm 0.35$ | $89.21 \pm 0.35$ | $90.40 \pm 0.33$ | $89.69 \pm 0.35$ | $88.98 \pm 0.36$ |
| | Size | $6.38 \pm 0.02$ | $3.89 \pm 0.02$ | $2.77 \pm 0.02$ | $3.89 \pm 0.03$ | $3.19 \pm 0.01$ | $2.10 \pm 0.02$ | $\underline{2.04 \pm 0.02}$ |
| | SSCV | $0.03 \pm 0.01$ | $\mathbf{0.03 \pm 0.01}$ | $0.05 \pm 0.01$ | $\mathbf{0.05 \pm 0.01}$ | $0.09 \pm 0.09$ | $0.03 \pm 0.01$ | $\underline{0.03 \pm 0.01}$ |
| Square | Coverage (%) | $89.44 \pm 0.36$ | $89.40 \pm 0.35$ | $90.86 \pm 0.32$ | $89.41 \pm 0.35$ | $88.24 \pm 0.37$ | $89.52 \pm 0.34$ | $89.09 \pm 0.34$ |
| | Size | $5.38 \pm 0.03$ | $4.16 \pm 0.02$ | $2.16 \pm 0.02$ | $3.10 \pm 0.02$ | $7.45 \pm 0.03$ | $1.96 \pm 0.02$ | $\underline{1.90 \pm 0.03}$ |
| | SSCV | $\underline{0.01 \pm 0.01}$ | $0.02 \pm 0.01$ | $\mathbf{0.06 \pm 0.01}$ | $0.01 \pm 0.03$ | $0.32 \pm 0.08$ | $0.05 \pm 0.01$ | $0.03 \pm 0.01$ |
| APGD[100] | Coverage (%) | $89.25 \pm 0.34$ | $89.40 \pm 0.35$ | $89.58 \pm 0.35$ | $89.59 \pm 0.33$ | $90.64 \pm 0.33$ | $89.81 \pm 0.34$ | $89.78 \pm 0.35$ |
| | Size | $5.76 \pm 0.03$ | $4.16 \pm 0.02$ | $2.64 \pm 0.02$ | $3.94 \pm 0.03$ | $7.66 \pm 0.03$ | $2.42 \pm 0.02$ | $\underline{2.32 \pm 0.20}$ |
| | SSCV | $\underline{0.02 \pm 0.01}$ | $0.02 \pm 0.01$ | $0.05 \pm 0.01$ | $0.05 \pm 0.01$ | $0.25 \pm 0.06$ | $0.05 \pm 0.00$ | $0.06 \pm 0.00$ |
| Auto | Coverage (%) | $89.22 \pm 0.36$ | $89.40 \pm 0.34$ | $89.58 \pm 0.35$ | $89.56 \pm 0.34$ | $88.24 \pm 0.36$ | $89.82 \pm 0.33$ | $89.76 \pm 0.34$ |
| | Size | $5.77 \pm 0.03$ | $4.16 \pm 0.02$ | $2.66 \pm 0.02$ | $3.95 \pm 0.03$ | $\mathbf{7.71 \pm 0.03}$ | $2.44 \pm 0.02$ | $\underline{2.36 \pm 0.02}$ |
| | SSCV | $\underline{0.02 \pm 0.01}$ | $0.02 \pm 0.01$ | $0.05 \pm 0.00$ | $0.05 \pm 0.01$ | $\mathbf{0.60 \pm 0.30}$ | $0.05 \pm 0.00$ | $0.06 \pm 0.00$ |
| OPSA[10] | Coverage (%) | $89.61 \pm 0.36$ | $89.11 \pm 0.34$ | $89.62 \pm 0.35$ | $90.46 \pm 0.32$ | $89.92 \pm 0.35$ | $89.70 \pm 0.33$ | $89.50 \pm 0.34$ |
| | Size | $\mathbf{7.29 \pm 0.02}$ | $\mathbf{4.50 \pm 0.02}$ | $\mathbf{4.02 \pm 0.02}$ | $\mathbf{5.40 \pm 0.02}$ | $7.34 \pm 0.02$ | $3.37 \pm 0.03$ | $\underline{3.11 \pm 0.03}$ |
| | SSCV | $\mathbf{0.09 \pm 0.00}$ | $\mathbf{0.04 \pm 0.01}$ | $0.03 \pm 0.01$ | $0.04 \pm 0.01$ | $\underline{0.02 \pm 0.01}$ | $0.02 \pm 0.00$ | $0.03 \pm 0.01$ |

*Table 1.* Mean and Standard Deviation of Coverage, Size, and SSCV for **CIFAR-10**

**Metrics.** To comprehensively assess the performance of shape-preserving prediction methods, we adopt three core metrics: *Coverage*, *Size*, and *Size-Stratified Coverage Violation* (SSCV).

The *Coverage* metric quantifies the proportion of test instances in $\mathcal{I}_{\text{test}}$ for which the true label is encompassed within the prediction set $\Gamma(\mathbf{x}; f, \tau)$, formulated as:

$$\text{Coverage} = \frac{1}{|\mathcal{I}_{\text{test}}|} \sum_{i \in \mathcal{I}_{\text{test}}} \mathbf{1}\left(y_i \in \Gamma(\mathbf{x}_i; f, \tau)\right). \quad (9)$$

A higher coverage value signifies that the prediction sets consistently include the true labels.

The *Size* metric assesses the average quantity of labels within the prediction sets across all test instances, given by:

$$\text{Size} = \frac{1}{|\mathcal{I}_{\text{test}}|} \sum_{i \in \mathcal{I}_{\text{test}}} |\Gamma(\mathbf{x}_i; f, \tau)|, \quad (10)$$

where smaller sizes imply more concise and informative predictions.

The *Size-Stratified Coverage Violation* (SSCV) (Angelopoulos et al., 2021) examines the consistency of coverage across varying prediction set sizes. It is defined as:

$$\text{SSCV}(\Gamma, \{S_j\}_{j=1}^s) =$$
$$\sup_{j \in [s]} \left| \frac{|i \in \mathcal{J}_j : y_i \in \Gamma(\mathbf{x}_i; f, \tau)|}{|\mathcal{J}_j|} - (1 - \alpha) \right|, \quad (11)$$

where $\{S_j\}_{j=1}^s$ partitions the possible prediction set sizes, and $\mathcal{J}_j = i \in \mathcal{I}_{\text{test}} : |\Gamma(\mathbf{x}_i; f, \tau)| \in S_j$. A smaller SSCV indicates more stable coverage across different set sizes.

Collectively, these metrics strike a balance between achieving the desired coverage probability and maintaining informative prediction sets, while ensuring that conformal prediction's coverage guarantees hold irrespective of the underlying model's accuracy.

**Results.** We present our experimentatal results in Tables 1 and 2. To facilitate comparison among attacks, we highlight the best results in each column in bold, while underlining the best defense results in each row. Following prior conformal prediction studies, we do not highlight results for *Coverage*, as they typically fluctuate around the theoretical $1 - \alpha$ level due to finite sample sizes and stochastic variability. Therefore, under the constraint of maintaining the

| Attacks | Indicator | Training Algorithm | | | | | |
|---------|-----------|------|------|------|------|------|------|
| | | **FGSM** | **PGD**[10] | **TRADES**[10] | **MART**[10] | **BETA-AT**[10] | **OPSA-AT**[10] |
| Clean | Coverage (%) | $90.26 \pm 0.32$ | $88.99 \pm 0.34$ | $90.50 \pm 0.33$ | $89.24 \pm 0.36$ | $91.39 \pm 0.31$ | $89.38 \pm 0.36$ |
| | Size | $13.47 \pm 0.01$ | $32.93 \pm 0.17$ | $8.81 \pm 0.07$ | $8.65 \pm 0.08$ | $73.65 \pm 0.27$ | $\underline{8.41 \pm 0.01}$ |
| | SSCV | $\mathbf{0.08 \pm 0.01}$ | $0.09 \pm 0.01$ | $0.07 \pm 0.01$ | $0.06 \pm 0.01$ | $0.40 \pm 0.28$ | $\underline{0.05 \pm 0.00}$ |
| FGSM | Coverage (%) | $90.67 \pm 0.33$ | $89.36 \pm 0.34$ | $90.55 \pm 0.33$ | $90.48 \pm 0.33$ | $88.50 \pm 0.37$ | $89.76 \pm 0.33$ |
| | Size | $60.36 \pm 0.21$ | $38.01 \pm 0.20$ | $28.27 \pm 0.17$ | $29.17 \pm 0.20$ | $25.71 \pm 0.12$ | $\underline{24.52 \pm 0.22}$ |
| | SSCV | $0.10 \pm 0.01$ | $0.08 \pm 0.01$ | $0.07 \pm 0.01$ | $0.06 \pm 0.01$ | $0.10 \pm 0.03$ | $\underline{0.03 \pm 0.01}$ |
| PGD[10] | Coverage (%) | $90.65 \pm 0.33$ | $89.31 \pm 0.34$ | $90.28 \pm 0.32$ | $90.74 \pm 0.34$ | $89.01 \pm 0.34$ | $89.58 \pm 0.34$ |
| | Size | $64.94 \pm 0.22$ | $38.22 \pm 0.20$ | $\underline{32.05 \pm 0.20}$ | $35.17 \pm 0.23$ | $68.70 \pm 0.22$ | $33.85 \pm 0.30$ |
| | SSCV | $0.10 \pm 0.00$ | $0.09 \pm 0.01$ | $0.06 \pm 0.01$ | $0.05 \pm 0.01$ | $\underline{0.03 \pm 0.03}$ | $0.04 \pm 0.01$ |
| PGD[40] | Coverage (%) | $90.66 \pm 0.32$ | $89.31 \pm 0.35$ | $90.25 \pm 0.34$ | $90.75 \pm 0.32$ | $89.25 \pm 0.34$ | $90.18 \pm 0.33$ |
| | Size | $64.95 \pm 0.34$ | $38.22 \pm 0.20$ | $\underline{32.04 \pm 0.19}$ | $35.14 \pm 0.24$ | $71.12 \pm 0.23$ | $\mathbf{33.74 \pm 0.28}$ |
| | SSCV | $0.10 \pm 0.00$ | $0.09 \pm 0.01$ | $0.07 \pm 0.01$ | $0.05 \pm 0.01$ | $0.07 \pm 0.05$ | $\underline{0.04 \pm 0.01}$ |
| BETA[10] | Coverage (%) | $90.49 \pm 0.33$ | $89.49 \pm 0.33$ | $90.36 \pm 0.32$ | $90.64 \pm 0.32$ | $88.36 \pm 0.35$ | $89.41 \pm 0.34$ |
| | Size | $57.59 \pm 0.22$ | $32.98 \pm 0.19$ | $20.38 \pm 0.14$ | $20.40 \pm 0.16$ | $23.44 \pm 0.11$ | $\underline{17.00 \pm 0.15}$ |
| | SSCV | $0.10 \pm 0.00$ | $0.09 \pm 0.01$ | $\mathbf{0.10 \pm 0.00}$ | $\mathbf{0.08 \pm 0.01}$ | $0.10 \pm 0.02$ | $\underline{0.06 \pm 0.02}$ |
| Square | Coverage (%) | $88.98 \pm 0.33$ | $88.99 \pm 0.35$ | $90.86 \pm 0.33$ | $89.79 \pm 0.34$ | $91.39 \pm 0.32$ | $89.31 \pm 0.34$ |
| | Size | $28.15 \pm 0.18$ | $33.03 \pm 0.18$ | $11.61 \pm 0.09$ | $\underline{11.60 \pm 0.10}$ | $74.69 \pm 0.27$ | $13.81 \pm 0.27$ |
| | SSCV | $0.07 \pm 0.01$ | $0.09 \pm 0.01$ | $0.09 \pm 0.00$ | $0.07 \pm 0.01$ | $0.40 \pm 0.26$ | $\underline{0.06 \pm 0.01}$ |
| APGD[100] | Coverage (%) | $89.05 \pm 0.35$ | $88.99 \pm 0.34$ | $90.22 \pm 0.34$ | $89.11 \pm 0.36$ | $91.37 \pm 0.31$ | $90.04 \pm 0.34$ |
| | Size | $32.00 \pm 0.21$ | $33.06 \pm 0.18$ | $13.54 \pm 0.11$ | $\underline{13.48 \pm 0.12}$ | $75.05 \pm 0.28$ | $16.01 \pm 0.20$ |
| | SSCV | $0.07 \pm 0.01$ | $0.09 \pm 0.01$ | $0.08 \pm 0.00$ | $0.07 \pm 0.01$ | $\mathbf{0.90 \pm 0.30}$ | $\underline{0.05 \pm 0.01}$ |
| Auto | Coverage (%) | $89.00 \pm 0.36$ | $88.99 \pm 0.36$ | $90.22 \pm 0.34$ | $89.12 \pm 0.36$ | $91.39 \pm 0.32$ | $89.50 \pm 0.31$ |
| | Size | $32.00 \pm 0.21$ | $33.06 \pm 0.18$ | $13.70 \pm 0.22$ | $13.58 \pm 0.12$ | $75.10 \pm 0.27$ | $\underline{13.42 \pm 0.16}$ |
| | SSCV | $0.07 \pm 0.01$ | $\mathbf{0.09 \pm 0.01}$ | $0.08 \pm 0.01$ | $0.07 \pm 0.01$ | $0.90 \pm 0.29$ | $\underline{0.05 \pm 0.01}$ |
| OPSA[10] | Coverage (%) | $90.81 \pm 0.32$ | $89.79 \pm 0.34$ | $90.31 \pm 0.33$ | $90.46 \pm 0.33$ | $89.36 \pm 0.36$ | $89.16 \pm 0.35$ |
| | Size | $\mathbf{65.56 \pm 0.22}$ | $\mathbf{39.23 \pm 0.20}$ | $\mathbf{32.30 \pm 0.19}$ | $\mathbf{35.38 \pm 0.23}$ | $\mathbf{75.25 \pm 0.20}$ | $\underline{32.07 \pm 0.25}$ |
| | SSCV | $\mathbf{0.10 \pm 0.00}$ | $0.07 \pm 0.01$ | $0.09 \pm 0.01$ | $0.07 \pm 0.01$ | $0.10 \pm 0.00$ | $\underline{0.03 \pm 0.01}$ |

*Table 2.* Mean and Standard Deviation of Coverage, Size, and SSCV for **CIFAR-100**

coverage guarantee, the primary metric of interest is the size of the prediction intervals.

As shown in Table 1, we conducted an additional experiment on CIFAR-10 using soft thresholding (the conformal training method proposed by Stutz et al. (2021)), denoted as OPSA-ST. Training with a mini-batch size of 64, our defense model demonstrates superior performance across all attack methods, yielding the minimal uncertainty (i.e., size) in the results. Furthermore, when excluding the two models inherently trained with OPSA attacks (OPSA-ST and OPSA-AT) and the BETA-AT defense model, our attack method consistently achieves the largest size among other defense frameworks. Notably, despite BETA-AT employing a training attack strategy similar to ours and OPSA-ST(including OPSA-AT) utilizing the OPSA attack methodology, the size produced by our attack method remains highly competitive with the strongest baseline attacks. To visually demonstrate these discrepancies, we present Box-violin plots (Figures

1-2) that illustrate the variance in attack performance across different defense models under identical experimental conditions.

Additionally, our experimental results on the CIFAR-100 dataset (Table 2) further validate the superiority of our defense framework and the potency of our attack methodology compared to existing approaches. Notably, our attack consistently produces the largest size across all evaluated defense models except when applied to our own proposed defense mechanism. Furthermore, our defense exhibits significantly reduced predictive uncertainty compared to baseline methods. Due to hardware limitations, we employed a batch size of 64 for standard training and a mini-batch size of 200 for conformal training experiments. While computational constraints currently restrict our implementation to these parameters, we included a remark (2) explaining that theoretical analysis suggests optimal performance would be achieved with mini-batch sizes between $500 - 1000$, as

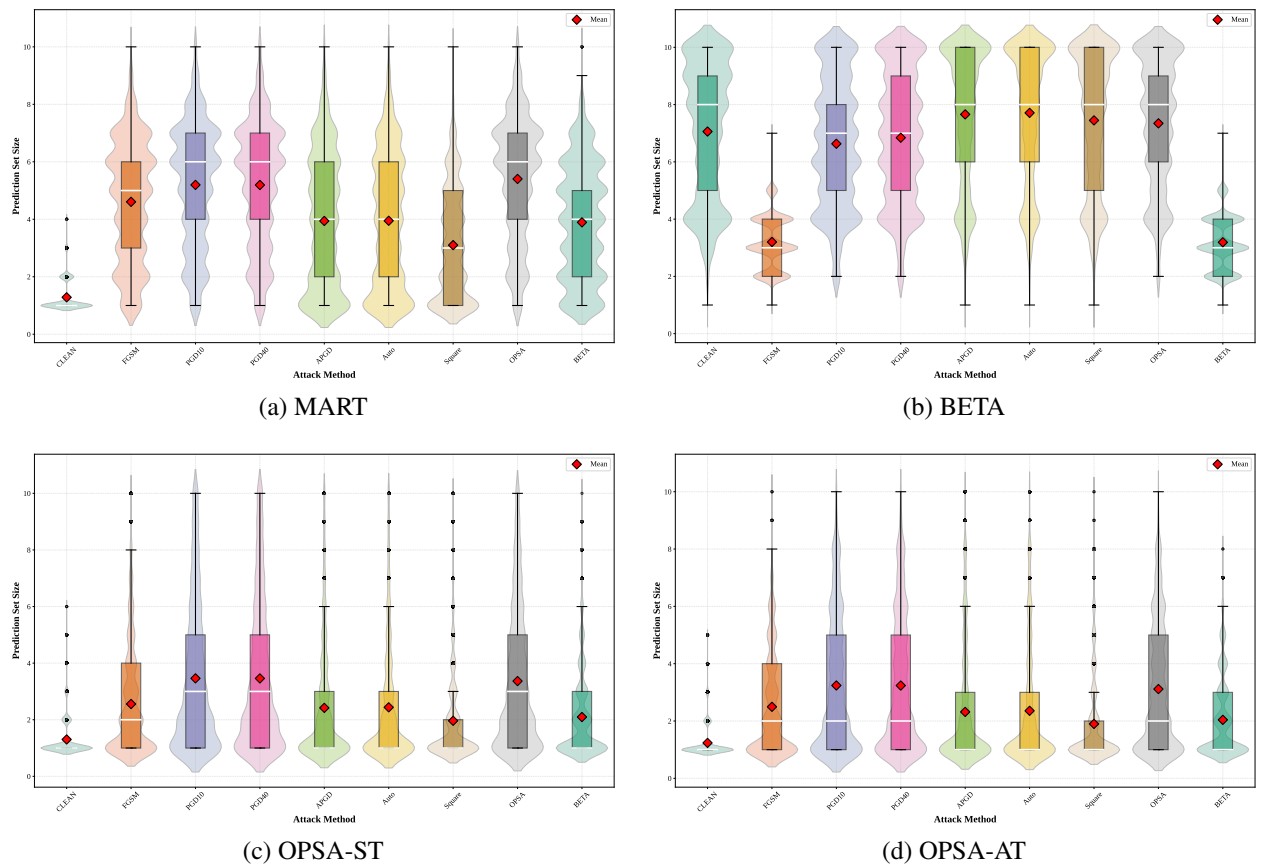

(a) MART

(b) BETA

(c) OPSA-ST

(d) OPSA-AT

*Figure 2.* Box-violin plots of CIFAR-10 results under MART, BETA-ST, OPSA-ST, and OPSA-AT defense models

this range better approximates the idealized exchangeability conditions required for robust conformal prediction.

*Remark* 2 (mini-Batch). In Section 3.4, we emphasized the critical importance of partitioning datasets into training ($\mathcal{B}_{\text{train}}$) and calibration ($\mathcal{B}_{\text{cal}}$) subsets during model training. To effectively approximate the THR procedure, maintaining exchangeability between $\mathcal{B}_{\text{train}}$ and $\mathcal{B}_{\text{cal}}$ within each mini-batch is essential. However, practical considerations of computational efficiency and memory constraints necessitate a balance. We recommend setting the subset size between $5 - 10$ times the number of classification categories ($K$) to optimize both approximation fidelity and training efficiency.

We further elaborate on supplementary experimental details across multiple appendices: Appendix B comprehensively profiles the computational overhead of various attack methodologies and defense models; Appendix C details our experimental results on the mini-ImageNet dataset; Appendix D presents comprehensive accuracy metrics for all evaluated defense models under diverse attack scenarios; and Appendix E provides an ablation study analyzing the impact of hyperparameter $T_1$ on experimental outcomes.

## 5. Conclusion

In this study, we introduce a novel framework that integrates adversarial training with conformal prediction to enhance the robustness of deep learning models against adversarial attacks. We treat adversarial training within this conformal framework as a dual-objective optimization challenge: on the one hand, our designed attack method aims to maximize the uncertainty of the prediction set without prior knowledge of the coverage rate; on the other hand, our defense method strives to minimize the uncertainty of the prediction set while maintaining a certain coverage rate. However, it's worth noting that our current experimental setup is relatively limited. We plan to expand our experiments by including the PreAct ResNet network in future studies. Experimental validations on CIFAR-10, CIFAR-100, and Mini-ImageNet datasets reveal that, compared to existing methods, our proposed attack method generates greater uncertainty, while the defense model demonstrates significantly improved robustness against various adversarial attacks. These findings strongly affirm the effectiveness of combining adversarial training with conformal prediction, providing new insights for developing reliable and resilient deep learning models in safety-critical applications.

## Acknowledgements

This work was partially supported by Hong Kong RGC and City University of Hong Kong grants (Project No. 9610639 and 6000864), Chengdu Municipal Office of Philosophy and Social Science grant 2024BS013, DFG grant No. 389792660, and VolkswagenStiftung Grant AZ 98514. Zhixin Zhou's research was supported by the Genesis Award for Scientific Breakthrough from Alpha Benito LLC.

## Impact Statement

The work presented in this paper aims to advance the field of machine learning, particularly through supplementary theoretical developments and explorations of adversarial training within the Conformal Prediction framework. There are many potential societal consequences of our work, none which we feel must be specifically highlighted here.

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

# A. Illustrative Example

To better understand the application of the improved negative margin in adversarial attacks within the conformal prediction framework, we use a one-dimensional prediction problem as an example. Please note that the parameter ranges here may differ from those in image-based scenarios, but this example sufficiently demonstrates the robustness of our algorithm. Consider the following example:

Assume a four-class classification problem with classes $K = \{1, 2, 3, 4\}$. The classifier $f$ outputs logits as follows:

$$f(x) = \begin{bmatrix} f_1(x) \\ f_2(x) \\ f_3(x) \\ f_4(x) \end{bmatrix} = \begin{bmatrix} 5 - x_1 \\ 3 + x_1 \\ 2 + 0.5x_1 \\ 1 - 0.2x_1 \end{bmatrix}.$$

The true class label is $y = 1$. The goal is to design an adversarial perturbation $\epsilon$ such that, when added to the input $\mathbf{x}$, it causes the prediction set $\Gamma(\mathbf{x}; f, \tau)$ to include more non-true classes, thereby increasing the model's uncertainty.

## A.1. Original Input

Consider the original input $\mathbf{x} = 2$. The logits are:

$$f(2) = \begin{bmatrix} 5 - 2 = 3 \\ 3 + 2 = 5 \\ 2 + 0.5 \times 2 = 3 \\ 1 - 0.2 \times 2 = 0.6 \end{bmatrix}.$$

Based on these logits, the prediction set $\Gamma(2)$ might be $\{1, 2\}$, assuming a confidence threshold that includes classes with logits close to the highest logit.

## A.2. Designing the Adversarial Perturbation

### A.2.1. STEP 1: DEFINE NEGATIVE MARGINS

For each class $k \in [K]$, define the negative margin as:

$$f_j(\mathbf{x} + \epsilon) - f_y(\mathbf{x} + \epsilon).$$

Specifically:

$$f_1(\mathbf{x} + \epsilon) - y = 0,$$

$$f_2(\mathbf{x} + \epsilon) - y = (3 + (x_1 + \epsilon)) - (5 - (x_1 + \epsilon)) = 2\epsilon + 2x_1 - 2,$$

$$f_3(\mathbf{x} + \epsilon) - y = (2 + 0.5(x_1 + \epsilon)) - (5 - (x_1 + \epsilon)) = 1.5\epsilon + x_1 - 3,$$

$$f_4(\mathbf{x} + \epsilon) - y = (1 - 0.2(x_1 + \epsilon)) - (5 - (x_1 + \epsilon)) = -0.2\epsilon - 0.2x_1 - 4.$$

### A.2.2. STEP 2: TEMPERATURE-SCALED NEGATIVE MARGIN

Introduce a temperature parameter $T = 1$ to scale the negative margins within the Sigmoid function:

$$M_T(\mathbf{x} + \epsilon; f, y) = \sum_{k \in [K]} \sigma \left( \frac{f_k(\mathbf{x} + \epsilon) - y}{T} \right) = \sigma(2\epsilon + 2x_1 - 2) + \sigma(1.5\epsilon + x_1 - 3) + \sigma(-0.2\epsilon - 0.2x_1 - 4).$$

### A.2.3. STEP 3: OPTIMIZATION OBJECTIVE

Formulate the adversarial perturbation $\epsilon^*$ as:

$$\epsilon^* = \arg\max_{\|\epsilon\|_\infty \le 0.5} \left[ \sigma(2\epsilon + 2x_1 - 2) + \sigma(1.5\epsilon + x_1 - 3) + \sigma(-0.2\epsilon - 0.2x_1 - 4) \right].$$

Given $\mathbf{x} = 2$:

$$M_T(\mathbf{x} + \epsilon; f, y) = \sigma(2\epsilon + 4 - 2) + \sigma(1.5\epsilon + 2 - 3) + \sigma(-0.2\epsilon - 0.4 - 4),$$

$$M_T(\mathbf{x} + \epsilon; f, y) = \sigma(2\epsilon + 2) + \sigma(1.5\epsilon - 1) + \sigma(-0.2\epsilon - 4.4).$$

### A.2.4. STEP 4: EVALUATING $M_T(\mathbf{x} + \epsilon; f, y)$

Evaluate $M_T(\mathbf{x} + \epsilon; f, y)$ for different $\epsilon$ values within the allowed range $[-0.5, 0.5]$:

| $\epsilon$ | $2\epsilon + 2$ | $1.5\epsilon - 1$ | $-0.2\epsilon - 4.4$ | $\sigma(2\epsilon + 2)$ | $\sigma(1.5\epsilon - 1)$ | $\sigma(-0.2\epsilon - 4.4)$ | $M_T(\mathbf{x} + \epsilon; f, y)$ |
|---|---|---|---|---|---|---|---|
| -0.5 | 1 | -1.75 | -4.3 | 0.7311 | 0.1521 | 0.0133 | 0.8965 |
| -0.25 | 1.5 | -1.375 | -4.35 | 0.8176 | 0.1839 | 0.0114 | 1.0129 |
| 0 | 2 | -1 | -4.4 | 0.8808 | 0.2689 | 0.0123 | 1.1619 |
| 0.25 | 2.5 | -0.625 | -4.45 | 0.9241 | 0.3446 | 0.0118 | 1.2805 |
| 0.3 | 2.6 | -0.55 | -4.46 | 0.9306 | 0.3685 | 0.0117 | 1.3108 |
| 0.4 | 2.8 | -0.4 | -4.48 | 0.9423 | 0.4013 | 0.0115 | 1.3541 |
| 0.5 | 3 | -0.25 | -4.5 | 0.9526 | 0.4378 | 0.0112 | 1.4016 |

*Table 3.* Evaluation of $M_T(\mathbf{x} + \epsilon; f, y)$ for different $\epsilon$ values.

### A.2.5. STEP 5: SELECTING OPTIMAL $\epsilon$

From the table, it is evident that $M_T(\mathbf{x} + \epsilon; f, y)$ increases as $\epsilon$ increases within the permissible range. Thus, the optimal perturbation is at the upper bound:

$$\epsilon^* = 0.5.$$

### A.2.6. STEP 6: IMPACT ON CONFORMAL PREDICTION

Applying $\epsilon^* = 0.5$ to the input $\mathbf{x} = 2$:

$$\mathbf{x} + \epsilon^* = 2 + 0.5 = 2.5.$$

Calculate the perturbed logits:

$$f(2.5) = \begin{bmatrix} 5 - 2.5 = 2.5 \\ 3 + 2.5 = 5.5 \\ 2 + 0.5 \times 2.5 = 3.25 \\ 1 - 0.2 \times 2.5 = 0.5 \end{bmatrix}.$$

Compute the negative margins:

$$M_1(2.5, 2.5, 1) = 0,$$

$$M(2.5, 5.5, 1)_2 = 5.5 - 2.5 = 3.0 > 0,$$

$$M(2.5, 3.25, 1)_3 = 3.25 - 2.5 = 0.75 > 0,$$

$$M'(2.5, 0.5, 1)_4 = 0.5 - 2.5 = -2.0 < 0.$$

| Dataset | Training Algorithm | | | | | |
|---|---|---|---|---|---|---|
| | FGSM | PGD[10] | TRADES[10] | MART[10] | BETA-AT[10] | OPSA-AT[10] |
| CIFAR-10 | 65 | 655 | 321 | 445 | 469 | 1642 |
| CIFAR-100 | 70 | 1398 | 342 | 742 | 1190 | 1689 |
| mini-ImageNet | 81 | 1766 | 452 | 881 | 1805 | 2024 |

*Table 4.* The time taken (in seconds) by each adversarial training model to complete one epoch of training on 100 batches, utilizing an NVIDIA A100 80GB GPU.

| Attacks | FGSM | PGD[10] | TRADES[10] | MART[10] | BETA-AT[10] | OPSA-AT[10] |
|---|---|---|---|---|---|---|
| FGSM | 18.5176 | 15.3600 | 17.2783 | 15.7824 | 18.5751 | 20.8580 |
| Auto | 1595.8604 | 2275.8080 | 2017.4780 | 2125.3660 | 669.0256 | 2543.7452 |
| Square | 272.9225 | 342.7212 | 336.6973 | 321.6511 | 107.9474 | 432.7322 |
| PGD[10] | 20.0746 | 18.3247 | 18.7224 | 19.7636 | 21.3279 | 21.5170 |
| PGD[40] | 27.1290 | 24.2051 | 24.0342 | 25.5516 | 29.1992 | 28.5025 |
| APGD[100] | 37.0082 | 39.8945 | 35.6490 | 35.7549 | 33.0879 | 35.6442 |
| BETA[10] | 32.7680 | 27.5900 | 27.1002 | 26.5268 | 29.0277 | 31.3223 |
| OPSA[10] | 25.9038 | 24.1116 | 25.4911 | 25.3905 | 27.2531 | 27.4187 |

*Table 5.* The attack execution time (in seconds) for various attack methods against different defense models, these results were derived from 8 batches of tests conducted on 100 images randomly sampled from the CIFAR-100 test set. Note that Auto-Attack was executed with its default parameters, and the square black-box attack was configured to perform 1000 queries.

Thus, the negative margins are:

$$M_T(\mathbf{x} + \epsilon; f, y) = \sigma(3.0) + \sigma(0.75) + \sigma(-2.0) \approx 0.9526 + 0.6792 + 0.1192 = 1.7509.$$

Since both $M(2.5, 5.5, 1)_2 > 0$ and $M(2.5, 3.25, 1)_3 > 0$, the prediction set $\Gamma(2.5)$ now includes classes 1, 2, and 3, i.e., $\Gamma(2.5) = \{1, 2, 3\}$.

**Impact Analysis:**

Before Perturbation: The prediction set $\Gamma(2) = \{1, 2\}$ included the true class 1 and one non-true class 2. After Perturbation: The prediction set $\Gamma(2.5) = \{1, 2, 3\}$ includes the true class 1 and two non-true classes 2 and 3.

This enlargement of the prediction set demonstrates that the adversarial perturbation successfully increases the model's uncertainty by incorporating additional non-true classes.

## B. Adversarial attack and adversarial training time

As detailed in this appendix, we present the training durations for various models and the computational overhead of different attack methods. Notably, our proposed model demonstrates scalability advantages - its training time does not escalate substantially with increasing dataset complexity, maintaining efficient performance across diverse experimental configurations.

## C. mini-ImageNet and figures

This appendix presents a comprehensive experimental evaluation on the mini-ImageNet dataset. As evidenced in Table 6 and Figure 3, the results clearly demonstrate both the superiority of our defense methodology and the remarkable efficacy of our attack strategy. Additionally, Figure 4 visualizes our experimental results on the CIFAR-100 dataset.

| Attacks | Indicator | Training Algorithm | | | | | |
|---------|-----------|------|------|------|------|------|------|
| | | **FGSM** | **PGD**[10] | **TRADES**[10] | **MART**[10] | **BETA-AT**[10] | **OPSA-AT**[10] |
| Clean | Coverage (%) | $90.38 \pm 0.27$ | $90.26 \pm 0.48$ | $89.17 \pm 0.51$ | $90.16 \pm 0.50$ | $90.08 \pm 0.47$ | $89.69 \pm 0.50$ |
| | Size | $43.52 \pm 0.70$ | $56.63 \pm 0.16$ | $10.91 \pm 0.10$ | $\underline{11.30 \pm 0.11}$ | $57.57 \pm 0.03$ | $17.02 \pm 0.17$ |
| | SSCV | $0.38 \pm 0.01$ | $\mathbf{0.90 \pm 0.20}$ | $0.09 \pm 0.01$ | $\underline{0.07 \pm 0.01}$ | $\mathbf{0.01 \pm 0.00}$ | $\mathbf{0.02 \pm 0.01}$ |
| FGSM | Coverage (%) | $90.49 \pm 0.47$ | $91.15 \pm 0.47$ | $90.49 \pm 0.50$ | $89.79 \pm 0.48$ | $89.90 \pm 0.46$ | $89.14 \pm 0.50$ |
| | Size | $49.49 \pm 0.12$ | $36.26 \pm 0.16$ | $38.66 \pm 0.17$ | $26.37 \pm 0.18$ | $57.51 \pm 0.01$ | $\underline{26.31 \pm 0.21}$ |
| | SSCV | $0.53 \pm 0.18$ | $0.08 \pm 0.10$ | $0.10 \pm 0.00$ | $\underline{0.15 \pm 0.06}$ | $0.01 \pm 0.00$ | $\underline{0.04 \pm 0.04}$ |
| PGD[10] | Coverage (%) | $89.97 \pm 0.50$ | $91.15 \pm 0.44$ | $90.89 \pm 0.47$ | $89.48 \pm 0.49$ | $89.64 \pm 0.49$ | $88.80 \pm 0.49$ |
| | Size | $52.55 \pm 0.12$ | $39.63 \pm 0.17$ | $43.19 \pm 0.17$ | $30.49 \pm 0.19$ | $57.31 \pm 0.01$ | $\underline{27.41 \pm 0.21}$ |
| | SSCV | $0.90 \pm 0.20$ | $0.10 \pm 0.01$ | $0.06 \pm 0.02$ | $\mathbf{0.23 \pm 0.12}$ | $0.01 \pm 0.00$ | $\underline{0.06 \pm 0.04}$ |
| PGD[40] | Coverage (%) | $89.92 \pm 0.49$ | $91.15 \pm 0.48$ | $90.89 \pm 0.48$ | $89.48 \pm 0.48$ | $89.71 \pm 0.49$ | $88.83 \pm 0.51$ |
| | Size | $52.59 \pm 0.49$ | $39.63 \pm 0.17$ | $43.19 \pm 0.18$ | $30.48 \pm 0.20$ | $57.31 \pm 0.01$ | $\underline{27.39 \pm 0.21}$ |
| | SSCV | $0.90 \pm 0.20$ | $0.10 \pm 0.01$ | $0.06 \pm 0.03$ | $0.23 \pm 0.12$ | $0.00 \pm 0.00$ | $\underline{0.06 \pm 0.04}$ |
| BETA[10] | Coverage (%) | $90.36 \pm 0.48$ | $91.07 \pm 0.44$ | $90.70 \pm 0.47$ | $90.29 \pm 0.49$ | $89.90 \pm 0.47$ | $88.88 \pm 0.49$ |
| | Size | $48.60 \pm 0.13$ | $35.63 \pm 0.16$ | $35.53 \pm 0.18$ | $21.64 \pm 0.15$ | $57.51 \pm 0.01$ | $\underline{25.62 \pm 0.21}$ |
| | SSCV | $0.40 \pm 0.16$ | $0.10 \pm 0.01$ | $0.10 \pm 0.00$ | $0.05 \pm 0.03$ | $0.00 \pm 0.00$ | $\underline{0.03 \pm 0.03}$ |
| Square | Coverage (%) | $90.98 \pm 0.33$ | $90.26 \pm 0.50$ | $90.57 \pm 0.46$ | $90.36 \pm 0.47$ | $90.08 \pm 0.49$ | $89.69 \pm 0.49$ |
| | Size | $50.24 \pm 0.18$ | $56.79 \pm 0.17$ | $21.59 \pm 0.17$ | $14.45 \pm 0.13$ | $57.57 \pm 0.03$ | $21.75 \pm 0.23$ |
| | SSCV | $0.57 \pm 0.01$ | $0.90 \pm 0.20$ | $0.08 \pm 0.01$ | $0.06 \pm 0.02$ | $0.00 \pm 0.00$ | $\underline{0.06 \pm 0.01}$ |
| APGD[100] | Coverage (%) | $90.05 \pm 0.35$ | $90.26 \pm 0.47$ | $90.05 \pm 0.49$ | $90.05 \pm 0.49$ | $90.08 \pm 0.00$ | $89.69 \pm 0.48$ |
| | Size | $51.37 \pm 0.21$ | $\mathbf{56.79 \pm 0.17}$ | $25.70 \pm 0.18$ | $16.24 \pm 0.14$ | $\mathbf{57.57 \pm 0.03}$ | $22.61 \pm 0.25$ |
| | SSCV | $0.50 \pm 0.01$ | $0.90 \pm 0.19$ | $0.08 \pm 0.02$ | $0.08 \pm 0.01$ | $0.00 \pm 0.00$ | $0.06 \pm 0.01$ |
| OPSA[10] | Coverage (%) | $90.16 \pm 0.48$ | $91.28 \pm 0.44$ | $90.31 \pm 0.33$ | $89.56 \pm 0.48$ | $89.92 \pm 0.49$ | $89.96 \pm 0.51$ |
| | Size | $\mathbf{53.42 \pm 0.11}$ | $40.00 \pm 0.16$ | $\mathbf{\underline{44.12 \pm 0.16}}$ | $\mathbf{30.84 \pm 0.19}$ | $57.50 \pm 0.00$ | $\underline{27.55 \pm 0.21}$ |
| | SSCV | $\mathbf{0.90 \pm 0.30}$ | $0.10 \pm 0.01$ | $\underline{0.08 \pm 0.02}$ | $0.16 \pm 0.09$ | $\underline{0.00 \pm 0.00}$ | $0.06 \pm 0.03$ |

*Table 6.* Mean and Standard Deviation of Coverage, Size, and SSCV for **mini-ImageNet**

## D. Accuracy metrics on various datasets

This appendix presents the accuracy of various defense models under different attack methods across three datasets, with Tables 7-9 summarizing the results for CIFAR-10, CIFAR-100, and mini-ImageNet respectively. Notably, while the OPSA attack method does not consistently yield the lowest accuracy, its focus on targeting uncertainty leads to significantly larger size metrics compared to alternative approaches.

## E. Parameter analysis

Regarding the hyperparameters $T_2$ and $\lambda$, we refer to the detailed analysis in (Stutz et al., 2021), where these parameters were rigorously analyzed. For $T_1$, its core function is to calibrate the sigmoid function to approximate the Threshold Response (THR) method (Sadinle et al., 2019), enlargement of prediction Interval. As shown in Table 10, systematic variations of on CIFAR-100 reveal a critical threshold effect: as randomly sampled values increase, the prediction set size expands progressively before plateauing at approximately 10.

| Indicator | Attacks | Training Algorithm | | | | | | |
|---|---|---|---|---|---|---|---|---|
| | | FGSM | PGD[10] | TRADES[10] | MART[10] | BETA-AT[10] | OPSA-ST[10] | OPSA-AT[10] |
| Accuracy | Clean | 72.84% | 48.48% | 77.84% | 81.69% | 55.65% | 81.33% | 89.15% |
| | FGSM | 36.11% | 44.32% | 54.81% | 43.35% | 37.04% | 64.75% | 65.83% |
| | PGD[10] | 29.60% | 43.85% | 50.76% | 25.36% | 29.06% | 53.60% | 57.27% |
| | PGD[40] | 29.60% | 43.86% | 50.75% | 34.36% | 28.94% | 53.43% | 57.13% |
| | BETA[10] | 39.75% | 49.64% | 63.80% | 50.09% | 43.79% | 70.57% | 70.34% |
| | Square | 27.56% | 43.09% | 54.54% | 40.29% | 33.04% | 58.26% | 61.04% |
| | APGD[100] | 24.75% | 42.16% | 51.37% | 33.16% | 12.21% | 52.05% | 56.07% |
| | Auto | 24.04% | 41.58% | 48.70% | 32.86% | 26.65% | 51.68% | 55.30% |
| | OPSA[10] | 30.21% | 44.39% | 51.58% | 35.09% | 19.48% | 54.16% | 57.91% |

*Table 7.* accuracy for **CIFAR-10**

| Indicator | Attacks | Training Algorithm | | | | | |
|---|---|---|---|---|---|---|---|
| | | FGSM | PGD[10] | TRADES[10] | MART[10] | BETA-AT[10] | OPSA-AT[10] |
| Accuracy | Clean | 43.65% | 24.15% | 51.55% | 52.10% | 6.07% | 55.65% |
| | FGSM | 16.10% | 20.69% | 30.23% | 30.18% | 24.24% | 37.04% |
| | PGD[10] | 12.85% | 20.49% | 26.70% | 25.35% | 5.29% | 29.06% |
| | PGD[40] | 12.81% | 20.47% | 26.67% | 25.27% | 4.93% | 28.94% |
| | BETA[10] | 17.94% | 24.69% | 38.22% | 39.77% | 25.94% | 43.79% |
| | Square | 10.87% | 20.05% | 28.99% | 28.68% | 2.38% | 33.04% |
| | APGD[100] | 9.88% | 19.46% | 25.47% | 24.32% | 1.19% | 27.73% |
| | Auto | 9.46% | 18.73% | 24.34% | 23.12% | 0.95% | 26.65% |
| | OPSA[10] | 13.14% | 20.89% | 27.60% | 26.27% | 5.78% | 29.99% |

*Table 8.* accuracy for **CIFAR-100**

| Indicator | Attacks | Training Algorithm | | | | | |
|---|---|---|---|---|---|---|---|
| | | FGSM | PGD[10] | TRADES[10] | MART[10] | BETA-AT[10] | OPSA-AT[10] |
| Accuracy | Clean | 18.39% | 4.71% | 43.67% | 45.13% | 1.56% | 55.65% |
| | FGSM | 9.61% | 20.47% | 15.42% | 25.39% | 24.24% | 37.04% |
| | PGD[10] | 5.78% | 16.72% | 10.73% | 20.70% | 1.56% | 29.06% |
| | PGD[40] | 5.70% | 16.72% | 10.65% | 25.27% | 1.56% | 28.94% |
| | BETA[10] | 10.86% | 21.17% | 19.14% | 20.73% | 1.56% | 43.79% |
| | Square | 3.26% | 2.19% | 28.99% | 22.14% | 1.56% | 33.04% |
| | APGD[100] | 2.06% | 2.08% | 16.24% | 24.32% | 1.56% | 27.73% |
| | Auto | 9.46% | 18.73% | 24.34% | 23.12% | 0.95% | 26.65% |
| | OPSA[10] | 6.07% | 17.14% | 11.15% | 21.38% | 1.56% | 29.99% |

*Table 9.* Accuracy for **mini-ImageNet**

| $T_1$ | OPSA | **OPSA-AT**[10] |
|---|---|---|
| 0.001 | Coverage (%) | $89.62 \pm 0.35$ |
| | Size | $22.06 \pm 0.20$ |
| | SSCV | $0.03 \pm 0.01$ |
| 0.1 | Coverage (%) | $89.76 \pm 0.34$ |
| | Size | $29.35 \pm 0.26$ |
| | SSCV | $0.03 \pm 0.01$ |
| 1 | Coverage (%) | $89.95 \pm 0.35$ |
| | Size | $33.30 \pm 0.27$ |
| | SSCV | $0.03 \pm 0.01$ |
| 10 | Coverage (%) | $89.88 \pm 0.34$ |
| | Size | $33.50 \pm 0.25$ |
| | SSCV | $0.03 \pm 0.01$ |
| 10 | Coverage (%) | $89.90 \pm 0.33$ |
| | Size | $33.50 \pm 0.26$ |
| | SSCV | $0.03 \pm 0.01$ |
| 1000 | Coverage (%) | $89.91 \pm 0.34$ |
| | Size | $33.50 \pm 0.26$ |
| | SSCV | $0.03 \pm 0.01$ |

*Table 10.* The effectiveness of OPSA attacks at different $T_1$.

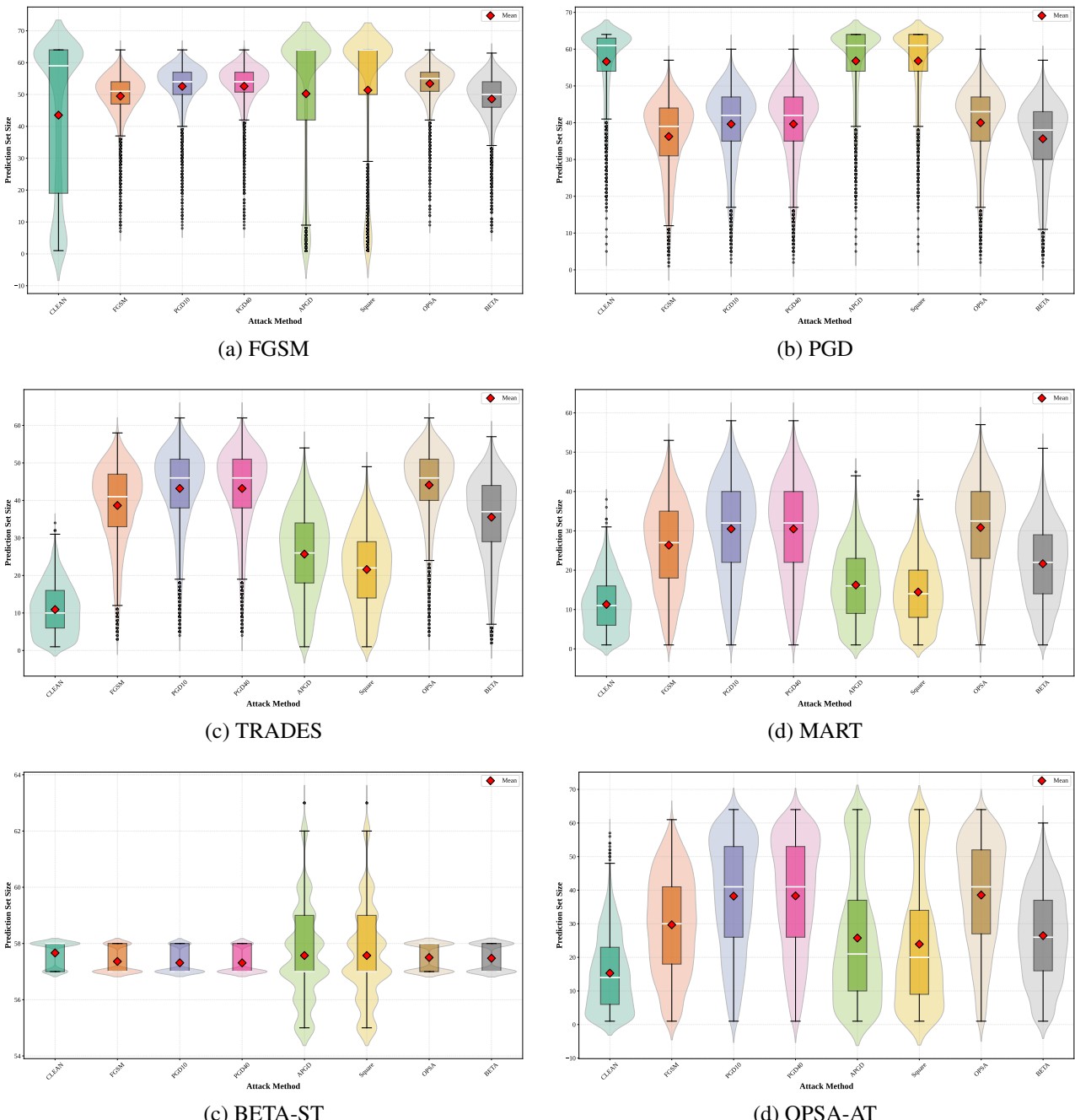

*Figure 3.* Box-violin plots of mini-ImageNet results under FGSM, PGD, TRADES, MART, BETA-ST, and OPSA-AT defense models

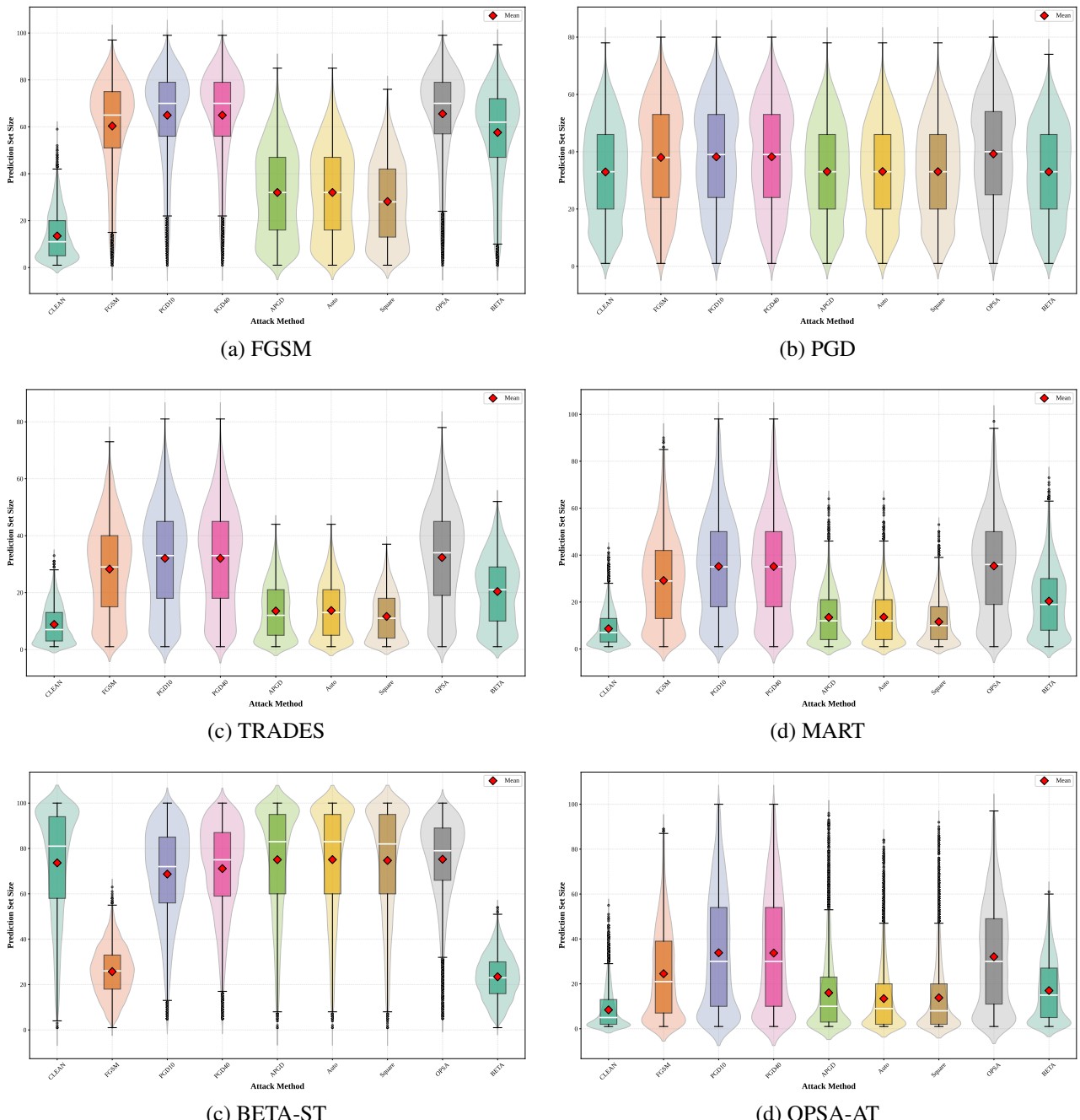

*Figure 4.* Box-violin plots of CIFAR100 results under FGSM, PGD, TRADES, MART, BETA-ST, and OPSA-AT defense models

