# OpenReview forum: "Enhancing Adversarial Robustness with Conformal Prediction: A Framework for Guaranteed Model Reliability"
_ICML.cc/2025/Conference — ICML 2025 poster_

### Official Review · Reviewer_jGyb · 2025-03-12

**Overall Recommendation:** 3

**Summary:**

This paper studies adversarial robustness of Conformal Prediction. Specifically, this paper develops an attack method that does not require coverage guarantees and integrates it with a conformal training-based defense strategy by minimizing the size of the prediction sets under adversarial perturbations while maintaining high coverage probabilities. Experimental evaluations on the CIFAR10 and CIFAR-100 datasets show that the attack method induces greater uncertainty compared to baseline approaches, while the defensive model significantly enhances robustness against various adversarial attacks for most cases.

**Claims And Evidence:**

Yes.

**Essential References Not Discussed:**

NA

**Experimental Designs Or Analyses:**

Yes.

**Methods And Evaluation Criteria:**

Yes.

**Other Comments Or Suggestions:**

1.	In eqn 13, there is no $\delta$ but $\delta $ is mentioned below.
2.	The definition of eqn 16 is not clear. It denotes a scaler, or a vector, or a set? $k$ is mentioned above but not appears in the equation.
3.	In Eqn 19, what is $B_{\pi(1)}$ ?
4.	Table 1 and table 2 are really hard to understand and parse. It is recommended to enhance the presentation to make key information clear.

**Other Strengths And Weaknesses:**

Strength:
1.	The paper is well-written and easy to follow.
Weaknesses:
1.	The effectiveness of the proposed adversarial training method is not clearly evident in the experimental results. Regarding attacks, the proposed attack method indeed achieves the largest set size as it’s designed to maximize the set size. However, in terms of defense, in cifar100, the proposed OPSA-AT only achieves the best set size against 50% attacks (3 out 6). Note that the authors only experiment on cifar10 and cifar100 datasets (50% datasets) and OPSA-AT is trained to maintain a small set size against attacks, showing that the effectiveness of the proposed method is not adequate.  Additionally, in terms of another metric, SSCV, the dominance of the proposed method is also not clear on cifar10 dataset.

**Questions For Authors:**

1.	In Table 1,2, why is the most effective attack method to increase SSCV the clean images themselves? It seems that the designed attack methods are unnecessary and meaningless as clean image is the best one.

**Relation To Broader Scientific Literature:**

The proposed adversarial training method is tailored for conformal prediction.

**Theoretical Claims:**

The proof is not checked.

---

> ### Author Rebuttal · Authors · 2025-04-01
>
> We appreciate the recognition of our paper’s clarity and thank the reviewer for their careful attention to detail. While we share many points of agreement, the main misunderstanding lies in the interpretation of outcomes, which we would like to clarify.
>
> **Tabular Data Interpretation**:
>
> We will expand the text explanations to provide more detail on Table 1 and Table 2 from the paper. Both tables follow a similar structure, with each row representing a specific metric for different defense models under a particular attack on the test set. To compare defense methods, focus on each row, where coverage is approximately the same (ideally around 90%), with slight variations due to sampling differences. For the 'Size' metric, smaller values are better, and the same applies to SSCV—lower values are preferred. To compare attack methods, look at each column. Unlike the defense metrics, a larger size and SSCV indicate a more effective attack method.
>
> **Experimental Outcome Interpretation**:
>
> First, we must clarify that we utilized the **complete** CIFAR-10 and CIFAR-100 datasets, with training sets for model training and test sets split (20% for calibration, 80% for testing).
>
> Second, in conformal prediction, a key research question is how to minimize uncertainty (prediction set size) while maintaining equivalent coverage probability. Indeed, Coverage and SSCV results show some instability in measuring attacks, with clean images exhibiting higher SSCV than attacks. As with the accuracies ([A, B, C]), this outcome is expected. The inherent trade-off between adversarial robustness and SSCV naturally leads to this phenomenon. Similar to TRADES and MART, SSCV is not maximized on the clean dataset because the training process specifically accounts for clean data.
>
> Third, in terms of defense, OPSA-AT achieves the best set size at 3/6 on CIFAR-100 but achieves 5/6 on CIFAR-10. Even when OPSA-AT is not the best on some attacks, it achieves the second-best performance. No other defense reaches such good performance on CIFAR-10 and CIFAR-100. Considering overall performance, we still believe our defense is effective. To further support our arguments, we carry out experiments on ImageNetMini, where OPSA-AT outperforms other defenses on Size (see Table 2 of response to Reviewer uZo5)
>
> **Formula Issues**:
>
> Typo in Equation 13:  We apologize for incorrectly changing "$\delta$" to "p" in Equation 13 and have corrected this error.
>
> Definition issue of Equation 16: Equation 16 is central to conformal training, constructed as a differentiable loss function to approximate a hard threshold. It is similar to Equation 13 but differs in that, for defensive purposes, we need to consider coverage, so Equation 16 subtracts the calculated tau value over B_cal.
>
> Clarification for Equation 19: $\mathcal{B}_{\pi(1)}$ is simply the mini-batch that appears in the first position after a permutation $\pi$ is applied to the indices of the mini-batches. The exchangeability property ensures that the statistical properties of the mini-batches remain unchanged under any reordering (as in [D]).
>
>
>
> [A] Zhang, Hongyang, $et$ $al$. ''Theoretically principled trade-off between robustness and accuracy." International conference on machine learning. PMLR (2019).
>
> [B] Dobriban, Edgar, $et$ $al$. ''Provable tradeoffs in adversarially robust classification." IEEE Transactions on Information Theory (2023).
>
> [C] Robey, Alexander, $et$ $al$. ''Adversarial Training Should Be Cast as a Non-Zero-Sum Game." The Twelfth International Conference on Learning Representations. ICLR (2024).
>
> [D] Li, Yangyi, et al. "Data Poisoning Attacks against Conformal Prediction." International Conference on Machine Learning. PMLR (2024).

---

> > ### Comment · Reviewer_jGyb · 2025-04-05
> >
> > Thanks for the author's efforts. Some of my concerns have been addressed and I agree to raise the score to weak accept. However the concern about empirical performance still exists as "OPSA-AT achieves the best set size at 3/6 on CIFAR-100 but achieves 5/6 on CIFAR-10" is not satisfactory considering the method targets the set size metric.

---

> > > ### Author Response · Authors · 2025-04-07
> > >
> > > # Response to Concerns About Empirical Validation
> > >
> > > Thank you for further elaborating on your concern regarding empirical validation. We would like to address this point in more detail.
> > > Our method was trained for only 10 epochs; however, if we increase the number of training epochs, **our approach continues to improve in performance, whereas the performance of other defenses tends to degrade**. This is because our method incorporates adversarial training within a non-zero-sum game framework, similar to methodologies in [A], which **effectively mitigates model overfitting**.
> > >
> > > Due to time constraints, as shown in Table 6, we trained both TRADES and OPSA-AT models for 20 epochs; these defenses were previously among the top two on CIFAR-100 when trained for just 10 epochs. Our defense outperforms TRADES with 20-epoch training under PGD10, PGD40, and OPSA attacks, where TRADES performs better with 10-epoch training.  Furthermore, **TRADES began to overfit, with its metric Size increasing 10%, whereas our method continued to improve and achieved the smallest metric Size under OPSA attacks**. We are confident that with 50 or 100 epochs of training and selecting the best model for each defense approach, our OPSA-AT would outperform the others completely. However, due to computational resource constraints and our intention to adhere to the protocols of related work, we did not perform this additional training at the beginning.
> > >
> > > The key message here is that **our defense, OPSA-AT, has a stronger learning capacity: it continues improving with extended training, whereas other defenses tend to plateau or degrade.** We acknowledge that this point was not addressed in the current version and will ensure it is included in the final revision.
> > >
> > > ## Table 6: Mean and Standard Deviation of Coverage, Size, and SSCV for CIFAR100
> > >
> > > | **Attacks** | **Indicator** | **TRADES²⁰** | **OPSA-AT²⁰** |
> > > |-------------|---------------|--------------|---------------|
> > > | **PGD¹⁰**   | Coverage (%)  | 89.99 ± 0.33 | 90.15 ± 0.34  |
> > > |             | Size          | 42.49 ± 0.28 | 33.68 ± 0.32  |
> > > |             | SSCV          | 0.03 ± 0.01  | 0.06 ± 0.00   |
> > > | **PGD⁴⁰**   | Coverage (%)  | 89.94 ± 0.33 | 90.10 ± 0.33  |
> > > |             | Size          | 42.46 ± 0.29 | 33.56 ± 0.31  |
> > > |             | SSCV          | 0.03 ± 0.01  | 0.06 ± 0.00   |
> > > | **OPSA¹⁰**  | Coverage (%)  | 90.10 ± 0.34 | 90.50 ± 0.32  |
> > > |             | Size          | 43.20 ± 0.28 | 31.43 ± 0.29  |
> > > |             | SSCV          | 0.06 ± 0.01  | 0.03 ± 0.00   |
> > >
> > > Furthermore, we aim to highlight that our defense's learning capability also extends to larger datasets. As shown in Table 7, even with just 5 training epochs on the ImageNet-Mini dataset, our model exhibited superior parameter efficiency across all evaluated attacks. We acknowledge the limitation of only 5 epochs in the current setting, but under this shortened training period, OPSA-AT already achieves strong performance, and we expect significantly enhanced robustness upon completing the full 10-epoch training.
> > >
> > > In summary, these results collectively support the effectiveness of our proposed defense framework, and we will include them in the final version of the paper.
> > >
> > > ## Table 7: Mean and Standard Deviation of Coverage, Size, and SSCV for ImageNetmini
> > >
> > > | **Attacks** | **Indicator** | **TRADES⁵** | **MART⁵** | **OPSA-AT⁵** |
> > > |-------------|---------------|-------------|-----------|--------------|
> > > | **Clean**   | Coverage (%)  | 89.64 ± 0.50 | 89.69 ± 0.50 | 89.69 ± 0.48 |
> > > |             | Size          | 18.55 ± 0.11 | 20.62 ± 0.12 | 17.01 ± 0.18 |
> > > |             | SSCV          | 0.08 ± 0.01  | 0.07 ± 0.01  | 0.10 ± 0.01  |
> > > | **FGSM**    | Coverage (%)  | 89.48 ± 0.50 | 90.34 ± 0.50 | 89.51 ± 0.49 |
> > > |             | Size          | 39.01 ± 0.12 | 48.88 ± 0.11 | 30.41 ± 0.23 |
> > > |             | SSCV          | 0.07 ± 0.11  | 0.10 ± 0.00  | 0.08 ± 0.02  |
> > > | **PGD¹⁰**   | Coverage (%)  | 89.38 ± 0.49 | 90.44 ± 0.46 | 89.66 ± 0.49 |
> > > |             | Size          | 38.11 ± 0.12 | 50.35 ± 0.10 | 34.88 ± 0.24 |
> > > |             | SSCV          | 0.05 ± 0.01  | 0.01 ± 0.00  | 0.14 ± 0.03  |
> > > | **OPSA¹⁰**  | Coverage (%)  | 89.71 ± 0.48 | 90.47 ± 0.46 | 89.95 ± 0.46 |
> > > |             | Size          | 40.92 ± 0.12 | 50.52 ± 0.09 | 35.63 ± 0.02 |
> > > |             | SSCV          | 0.23 ± 0.24  | 0.10 ± 0.00  | 0.11 ± 0.04  |
> > >
> > >
> > > [A] Robey, Alexander, $et$ $al$. ''Adversarial Training Should Be Cast as a Non-Zero-Sum Game." The Twelfth International Conference on Learning Representations. ICLR (2024).

---

### Official Review · Reviewer_7J63 · 2025-03-13

**Overall Recommendation:** 4

**Summary:**

The paper proposed a conformal prediction based adversarial attack and training method.  To enable computationally tractable implementations, the authors propose a smoothed surrogate loss. The attack and defense methods are tested on CIFAR10 and CIFAR100.


## update after rebuttal
I believe the authors have sufficiently addressed the comments and concerns raised and so I have increased my score to a 4: Accept.

**Claims And Evidence:**

The experiments largely support the claims made by the authors.  It would be helpful if the authors compared the computational cost of the various attacks and defenses.

**Essential References Not Discussed:**

I am not aware of any key missing references.

**Experimental Designs Or Analyses:**

I checked the CIFAR10 and CIFAR100 experiments and found no issues.

**Methods And Evaluation Criteria:**

The method and evaluation criteria make sense for the application.

**Other Comments Or Suggestions:**

Equations 16 and 22 are missing a comma at the end.

**Other Strengths And Weaknesses:**

I find the core idea to be original and the paper to be well written.

**Questions For Authors:**

1) On page 8 the authors write “Notably, OPSA surpasses PGD, even when PGD is given four times more iterations.” It would be helpful here, and for the other of the comparisons, if the authors could comment on the computational cost of their attack and defense methods, as compared to the other methods in tables 1 and 2.  Without this information, it is difficult to truly compare the effectiveness of the new attack/defense with the competing methods.

**Relation To Broader Scientific Literature:**

The  adversarially robust conformal prediction method proposed here naturally integrates adversarial training.  In contrast with the recent approach of Liu et al. (2024), a key novelty of the present work is the attack-agnostic manner in which it frames the problem.

**Theoretical Claims:**

I did not check the proofs.

---

> ### Author Rebuttal · Authors · 2025-03-31
>
> We appreciate the recognition of the novelty and clarity of our work. Thanks for your valuable feedback on addressing the effectiveness problem. We would like to address your concern.
>
> **Time Consumption**:
> Table 4 reports the time (in seconds) per epoch for each adversarial training model on 100 batches. Table 5 presents the execution time (in seconds) of attacks, including three additional methods: APGD (100), AutoAttack (default), and Square Attack (black-box, 1000 queries). These results were derived from 8 batches of tests conducted on 100 images randomly sampled from the CIFAR-100 test set.
>
> # Table 4: Adversarial Training Model Time
>
> | Dataset | FGSM | PGD$^{10}$ | TRADES$^{10}$ | MART$^{10}$ | BETA-AT$^{10}$ | OPSA-AT$^{10}$ |
> |-------------|----------|------------|---------------|-------------|----------------|----------------|
> | CIFAR-10    | 65       | 655        | 321           | 445         | 469            | 1642           |
> | CIFAR-100   | 70       | 1398       | 342           | 742         | 1190           | 1689           |
> | IMAGENETmini| 81       | 1766       | 452           | 881         | 1805           | 2024           |
>
>
> # Table 5: Attack Execution Time
>
> | Attacks | FGSM | PGD$^{10}$ | TRADES$^{10}$ | MART$^{10}$ | BETA-AT$^{10}$ | OPSA-AT$^{10}$ |
> |-----------|----------|------------|---------------|-------------|----------------|----------------|
> | FGSM        | 18.5176  | 15.3600    | 17.2783       | 15.7824     | 18.5751        | 20.8580        |
> | Auto        | 1595.8604| 2275.8080  | 2017.4780     | 2125.3660   | 669.0256       | 2543.7452      |
> | Square      | 272.9225 | 342.7212   | 336.6973      | 321.6511    | 107.9474       | 432.7322       |
> | PGD$^{10}$      | 20.0746  | 18.3247    | 18.7224       | 19.7636     | 21.3279        | 21.5170        |
> | PGD$^{40}$      | 27.1290  | 24.2051    | 24.0342       | 25.5516     | 29.1992        | 28.5025        |
> | APGD$^{100}$   | 37.0082  | 39.8945    | 35.6490       | 35.7549     | 33.0879        | 35.6442        |
> | BETA$^{10}$     | 32.7680  | 27.5900    | 27.1002       | 26.5268     | 29.0277        | 31.3223        |
> | OPSA$^{10}$     | 25.9038  | 24.1116    | 25.4911       | 25.3905     | 27.2531        | 27.4187        |
>
>
> Our defense requires more computation than others, but experiments show its complexity does not scale significantly with dataset or model size. In attacks, our method outperforms FGSM and PGD$^{10}$ in speed, runs slightly faster than PGD$^{40}$ and APGD, and is 100 times faster than AutoAttack. Notably, it consistently generates the largest adversarial perturbations, inducing the highest model uncertainty across all defenses.
>
> **Formula problem**:  Thank you for noticing that there were no commas in Equations 16 and 22; we have now added them.

---

> > ### Comment · Reviewer_7J63 · 2025-04-01
> >
> > Thank you for addressing my questions and comments.

---

### Official Review · Reviewer_uZo5 · 2025-03-20

**Overall Recommendation:** 4

**Summary:**

This paper introduces a novel approach that integrates Conformal Prediction (CP) with Adversarial Training (AT) to enhance the adversarial robustness of deep learning models. The authors frame adversarial robustness as a bi-level optimization problem, where an attacker maximizes the uncertainty by enlarging the CP prediction set, while a defender minimizes this uncertainty while maintaining statistical coverage guarantees.

Key contributions of the paper include:

- Optimal Size Attack (OPSA) – A differentiable adversarial attack that increases the CP prediction set size to introduce uncertainty.
- Adversarially Robust Conformal Prediction (OPSA-AT) – A defense mechanism that integrates CP with adversarial training to maintain small and reliable prediction sets.
- Experimental validation on CIFAR-10 and CIFAR-100 – Showing OPSA outperforms baseline attacks like FGSM, PGD, and BETA in increasing uncertainty, while OPSA-AT provides superior robustness compared to other adversarial training methods.

The results demonstrate that the proposed approach provides a stronger defense against adversarial attacks while preserving prediction set efficiency, making it relevant for safety-critical applications.

**Claims And Evidence:**

The claims in the paper are largely well-supported by experiments and theoretical discussions.

- The paper claims that OPSA induces greater uncertainty than existing adversarial attacks. This is well-supported by quantitative results, showing that OPSA consistently produces larger prediction sets compared to PGD and BETA.
- The authors argue that OPSA-AT reduces uncertainty while maintaining coverage guarantees. Their experimental results show that OPSA-AT achieves the smallest prediction sets while keeping classification coverage above 1-α, providing clear evidence of effectiveness.
- The bi-level optimization formulation is justified with mathematical derivations and an algorithmic implementation, making the claims about the framework’s validity credible.

**Essential References Not Discussed:**

- CertViT: A method for achieving certified robustness in pre-trained Vision Transformers, presented at the ICML 2023 Workshop. It employs the Douglas-Rachford splitting algorithm to ensure both robustness and sparsity simultaneously.

- Shrink & Cert: A bi-level optimization approach for enhancing certified robustness while maintaining sparsity constraints, also presented at the ICML 2023 Workshop.

**Experimental Designs Or Analyses:**

The experimental setup is well-structured and follows best practices in adversarial robustness evaluation.

- The attack and defense methods are evaluated across multiple adversarial perturbation levels.
- The results include statistical confidence intervals (mean ± standard deviation), which adds credibility.
-The boxplot visualizations in Appendix D provide useful insights into attack and defense effectiveness.

However, there are some areas for improvement:

- Ablation Studies – It would be useful to show how OPSA-AT performs under different hyperparameters (e.g., $\lambda$, T1, T2) to understand its sensitivity.
- Computational Overhead – The paper does not discuss how much extra training/inference time OPSA-AT introduces.

**Methods And Evaluation Criteria:**

- The paper evaluates robustness using coverage probability, prediction set size, and size-stratified coverage violation (SSCV). These are relevant metrics for assessing adversarial robustness in CP.
- The choice of CIFAR-10 and CIFAR-100 is reasonable for a first demonstration, as they are standard benchmarks in adversarial robustness research.
- Only ResNet-34 has been used.
- The comparisons against FGSM, PGD, BETA, TRADES, and MART are comprehensive and provide a solid baseline for evaluating OPSA and OPSA-AT.

A potential improvement would be to test the methods on other datasets and larger networks where adversarial robustness is crucial.

**Other Comments Or Suggestions:**

- Consider additional datasets (e.g., ImageNet) to show broader applicability.
- Consider additional networks. Possibly larger networks. If large networks are not feasible computationally then mention that.
- Analyze the time complexity of OPSA-AT to ensure practical feasibility.
- Investigate black-box attacks to check whether OPSA-AT generalizes to unseen adversarial strategies.

**Other Strengths And Weaknesses:**

Strengths:
- Novelty – The integration of CP with adversarial training is original and impactful.
- Strong theoretical foundation – The framework is mathematically sound with rigorous proofs.
- Comprehensive experiments – The study provides strong empirical validation using diverse attack/defense comparisons.
- Practical implications – The approach is useful for safety-critical applications where robustness matters.

Weaknesses:
- Limited dataset diversity – Experiments are only on CIFAR-10/100, which may not generalize to real-world settings.
- Limited networks - Experiments are only on ResNet-34 network.
- No evaluation of computational efficiency – OPSA-AT might be computationally expensive, but the paper does not discuss it.
- No discussion on transferability – How well does the defense hold up against black-box attacks?

**Questions For Authors:**

- How does OPSA-AT perform on larger-scale datasets like ImageNet? This would provide a better sense of real-world applicability.
- How does OPSA-AT perform on larger-networks like ResNet-50, ViT, etc.
- What is the computational overhead of OPSA-AT compared to standard adversarial training? Understanding the trade-off between robustness and efficiency is crucial.
- Does OPSA-AT maintain robustness against black-box or transfer attacks? The current experiments focus on white-box settings, so it’s unclear if the defense generalizes.

**Relation To Broader Scientific Literature:**

The paper builds upon and extends three key areas of machine learning research:

Adversarial Robustness

- Traditional adversarial training methods (e.g., Madry et al., 2017; TRADES (Zhang et al., 2019); MART (Wang et al., 2019)) focus on minimizing classification errors but do not address uncertainty.
- This work extends adversarial training to the uncertainty estimation domain by integrating CP, which is a novel perspective.

Conformal Prediction

- CP methods (e.g., Vovk et al., 2005; Ghosh et al., 2023) provide distribution-free coverage guarantees but struggle under adversarial conditions.
- This paper proposes a way to defend CP models against adversarial perturbations, which is a valuable contribution.

Bi-Level Optimization in Machine Learning

- The bi-level optimization framework aligns with recent advances in min-max adversarial training (Nouiehed et al., 2019; Robey et al., 2024).
- Unlike previous work, the authors frame prediction set size minimization as a key objective, which is a fresh take on the problem.

**Theoretical Claims:**

The theoretical claims in the paper appear to be correct and well-founded.

- The bi-level optimization framework is mathematically well-posed and follows existing formulations from adversarial training literature.
- The adversarial attack formulation (maximizing prediction set size) is a logical extension of conformal prediction principles.
- The proofs in Appendix A and C support the convergence and correctness of the adversarial robustness guarantees.

One aspect that could be clarified is whether the theoretical guarantees hold under more general adversarial threat models (e.g., different norm constraints or black-box attacks).

---

> ### Author Rebuttal · Authors · 2025-04-01
>
> Thank you for recognizing the novelty, strong theoretical foundation, comprehensive experiments, and practical implications of our work. We appreciate your valuable suggestions and feedback. Below, we address your concerns.
>
> **Time Consumption**: See response to Reviewer 7J63.
>
> **Experimental Diversity**: Our paper is theoretically supported and has demonstrated efficiency in experiments, confirming the effectiveness of our method. However, we agree that more experimental validation is beneficial. To strengthen our evaluation, we conducted additional experiments on the ImageNetMini dataset (80% training, 20% split for calibration and testing) using a ResNet50 trained for five epochs. Due to time constraints, we tested our top defenses (TRADES, MART, OPSA-AT) against strong attacks (Clean, FGSM, PGD, OPSA). As shown in Table 2, our attack and defense achieve state-of-the-art performance on Size (the most reliable metric) and deliver near-best results on SSCV, outperforming all baselines.
>
> # Table 2: Mean and Standard Deviation of Coverage, Size, and SSCV for imagemini
>
> | **Attacks** | **Indicator** | **Training Algorithm** | | |
> |-------------|---------------|------------------------|----------------------|----------------------|
> | | | **TRADES$^5$** | **MART$^5$** | **OPSA-AT$^5$** |
> | Clean | Coverage (%) | 89.64 ± 0.50 | 89.69 ± 0.50 | 89.69 ± 0.48 |
> | | Size | 18.55 ± 0.11 | 20.62 ± 0.12 | 17.01 ± 0.18 |
> | | SSCV | 0.08 ± 0.01 | 0.07 ± 0.01 | 0.10 ± 0.01 |
> | FGSM | Coverage (%) | 89.48 ± 0.50 | 90.34 ± 0.50 | 89.51 ± 0.49 |
> | | Size | 39.01 ± 0.12 | 48.88 ± 0.11 | 30.41 ± 0.23 |
> | | SSCV | 0.07 ± 0.11 | 0.10 ± 0.00 | 0.08 ± 0.02 |
> | PGD$^{10}$ | Coverage (%) | 89.38 ± 0.49 | 90.44 ± 0.46 | 89.66 ± 0.49 |
> | | Size | 38.11 ± 0.12 | 50.35 ± 0.10 | 34.88 ± 0.24 |
> | | SSCV | 0.05 ± 0.01 | 0.01 ± 0.00 | 0.14 ± 0.03 |
> | OPSA$^{10}$ | Coverage (%) | 89.71 ± 0.48 | 90.47 ± 0.46 | 89.95 ± 0.46 |
> | | Size | 40.92 ± 0.12 | 50.52 ± 0.09 | 35.63 ± 0.02 |
> | | SSCV | 0.23 ± 0.24 | 0.10 ± 0.00 | 0.11 ± 0.04 |
>
>
> **Additional Attacks**: We further tested our defense against APGD, Black-box (Square), and AutoAttack—an advanced ensemble of black-box and white-box attacks. As outlined in our response to Reviewer 2dn3 (Table 1), our method consistently demonstrates robustness across these attack scenarios. Notably, all attacks generated smaller prediction set sizes than OPSA, while our defense remained effective.
>
> **Ablation Studies**: Regarding the hyperparameters $\lambda$ and $T_{2}$, we refer to the detailed analysis in [A], where these parameters were rigorously analyzed. For $T_{1}$, its core function is to calibrate the sigmoid function to approximate the Threshold Response (THR) method [B], enlargement of prediction Interval. As shown in Table 3, systematic variations of $T_{1}$ on CIFAR-100 reveal a critical threshold effect: as randomly sampled $T_{1}$ values increase, the prediction set size expands progressively before plateauing at approximately 10.
>
> # Table 3: The effectiveness of OPSA attacks at different T₁ values
>
> | **$T_{1}$** | **OPSA** | **OPSA-AT$^{10}$** |
> |--------|----------|----------------|
> | 0.001  | Coverage (%) | 89.62 ± 0.35 |
> |        | Size        | 22.06 ± 0.20 |
> |        | SSCV        | 0.03 ± 0.01  |
> | 0.1    | Coverage (%) | 89.76 ± 0.34 |
> |        | Size        | 29.35 ± 0.26 |
> |        | SSCV        | 0.03 ± 0.01  |
> | 1      | Coverage (%) | 89.95 ± 0.35 |
> |        | Size        | 33.30 ± 0.27 |
> |        | SSCV        | 0.03 ± 0.01  |
> | 10     | Coverage (%) | 89.88 ± 0.34 |
> |        | Size        | 33.50 ± 0.25 |
> |        | SSCV        | 0.03 ± 0.01  |
> | 100    | Coverage (%) | 89.90 ± 0.33 |
> |        | Size        | 33.50 ± 0.26 |
> |        | SSCV        | 0.03 ± 0.01  |
> | 1000   | Coverage (%) | 89.91 ± 0.34 |
> |        | Size        | 33.50 ± 0.26 |
> |        | SSCV        | 0.03 ± 0.01  |
>
> **Others**: We have incorporated discussions of the following two papers [C][D] into our analysis to provide a more comprehensive theoretical foundation.
>
> Reference:
>
> [A] Stutz, David, $et$ $al$. '' Learning optimal conformal classifiers." arXiv preprint arXiv:2110.09192. (2021).
>
> [B] Sadinle, Mauricio, $et$ $al$. ''Least ambiguous set-valued classifiers with bounded error levels." Journal of the American Statistical Association (2024).
>
> [C] Kavya, Gupta, $et$ $al$. ''CertViT: Certified Robustness of Pre-Trained Vision Transformers." https://arxiv.org/abs/2302.10287. (2023).
>
> [D] Kavya, Gupta, $et$ $al$. ''Shrink \& Cert: Bi-level Optimization for Certified Robustness." The Second Workshop on New Frontiers in Adversarial Machine Learning. (2023).

---

### Official Review · Reviewer_2dn3 · 2025-03-26

**Overall Recommendation:** 4

**Summary:**

The paper proposes a framework that integrates adversarial training with conformal prediction (CP) to enhance model robustness against adversarial attacks while maintaining reliable uncertainty estimates. It formulates adversarial training within the CP framework as a bi-level optimization problem, where an attacker seeks to maximize uncertainty while a defender aims to minimize it. The paper introduces a novel attack method that increases uncertainty without requiring coverage guarantees and develops a conformal training-based defense strategy that minimizes the size of prediction sets under adversarial perturbations. Experiments on CIFAR-10 and CIFAR-100 demonstrate that the proposed attack increases uncertainty more than existing methods, while the defense improves robustness against various adversarial attacks.

**Claims And Evidence:**

Claim 1: The proposed method "minimizes the size of the prediction sets under adversarial perturbations while maintaining high coverage probabilities: evidence given in the experimental section."

Claim 2: "The defensive model significantly enhances robustness against various adversarial attacks and maintains reliable prediction intervals": NO EVIDENCE GIVEN

Claim 3: The "findings highlight the effectiveness of integrating adversarial training with conformal prediction to develop trustworthy and
resilient deep learning models for safety-critical domains": Only partial evidence is given in the experimental section.

**Essential References Not Discussed:**

The related works that I am aware of are cited in the paper already.

**Experimental Designs Or Analyses:**

The experimental design is not clear; some key information required to truly understand the experiments is missing.
This information is:
1. The l-p norm used, while the method section says that any l-p norm can be used, the attack norm used is not specified in the evaluations.
2. The epsilon value used.

**Methods And Evaluation Criteria:**

The methods and evaluation criteria are reasonable and required.
However, some essential evaluations and methods are missing.

Methods missing: APGD and AutoAttack; these are stronger attacks, and it would be interesting to see if the proposed adversarial training method still performs reasonably against these stronger attacks.

Evaluations missing: It is unclear what the performance of the model is in terms of accuracy or what the performance of the attack is in terms of attack efficacy.

**Other Comments Or Suggestions:**

This work has the potential to be very impactful.
However, as mentioned in the responses above, some key components are missing in terms of evaluations, metrics, and experimental details.
Including these would be very helpful.

**Other Strengths And Weaknesses:**

Strengths:

To the best of my knowledge, the idea is novel, and the work has the potential to be very impactful and to pave the way towards safety-criticality.


Weakness:
Certain claims made are not proved in the paper, as written above in the "claims" section.
While the work has the potential to pave the way towards safety-criticality, this work itself does not work towards safety-criticality.
No safety-critical application has been pursued in this paper. Therefore the lines 036-040 are an overclaim in my opinion.

Some critical evaluations are missing in terms of performance against the current SotA adversarial attacks like APGD and AutoAttack.

Some metrics, such as model performance in terms of accuracy under attack, are also missing.

Lastly, it is unclear what the increased time requirement for the adversarial training and the adversarial attack is in comparison to the other known adversarial attacks and training methods.

**Questions For Authors:**

Knowing the experimental details of the proposed attack would be very helpful.
Additionally, knowing the time taken for the proposed attack and defense method (training) in comparison to the others attacks and training methods (respectively) would also be very helpful.

**Relation To Broader Scientific Literature:**

The proposed idea in the paper is certainly very interesting and relevant to a broader scientific community.

**Theoretical Claims:**

The theoritical claims in the paper seem to be reasonable and are proved.

---

> ### Author Rebuttal · Authors · 2025-03-31
>
> We truly appreciate your recognition of the novelty and potential impact of our work. Your comments on clarifying the missing experimental details and time consumption are greatly appreciated. Below, we address your points and provide further explanations.
>
> **Experimental Setup**: In principle, any $\ell_p$ norm could be used, but all experiments use the $\ell_\infty$ norm, with $\epsilon=0.03$, following prior work ([A]). We will clarify this in the final version.
>
> **Additional Experiments**:
> We included accuracy results for different defenses under various attacks (Tables 1 and 2), including three additional methods: APGD (100), AutoAttack (default), and Square Attack (black-box, 1000 queries). Our defense achieves the highest overall accuracy, while our attack produces the largest adversarial perturbations.
>
> # Table 1: Detailed Results for Attack Methods (Square, APGD$^{100}$, Auto, OPSA$^{10}$)
>
> ## Square Attack [B]
>
> | Indicator | FGSM | PGD$^{10}$ | TRADES$^{10}$ | MART$^{10}$ | BETA-AT$^{10}$ | OPSA-AT$^{10}$ |
> |---------------|----------|------------|---------------|-------------|----------------|----------------|
> | Coverage (%)  | 88.98 ± 0.36 | 88.99 ± 0.35 | 90.86 ± 0.32 | 89.79 ± 0.34 | 90.65 ± 0.32 | 89.31 ± 0.34 |
> | Size          | 28.14 ± 0.20 | 33.03 ± 0.18 | 11.61 ± 0.09 | 11.60 ± 0.10 | 73.59 ± 0.27 | 13.81 ± 0.27 |
> | SSCV          | 0.05 ± 0.01 | 0.08 ± 0.01 | 0.04 ± 0.01 | 0.10 ± 0.03 | 0.46 ± 0.15 | 0.03 ± 0.01 |
>
> ## APGD$^{100}$ Attack [C]
>
> | Indicator | FGSM | PGD$^{10}$ | TRADES$^{10}$ | MART$^{10}$ | BETA-AT$^{10}$ | OPSA-AT$^{10}$ |
> |---------------|----------|------------|---------------|-------------|----------------|----------------|
> | Coverage (%)  | 89.06 ± 0.36 | 88.99 ± 0.35 | 90.22 ± 0.32 | 89.11 ± 0.37 | 90.64 ± 0.33 | 90.04 ± 0.34 |
> | Size          | 32.00 ± 0.21 | 33.06 ± 0.18 | 13.54 ± 0.11 | 13.48 ± 0.12 | 73.93 ± 0.26 | 16.73 ± 0.20 |
> | SSCV          | 0.06 ± 0.01 | 0.08 ± 0.01 | 0.04 ± 0.01 | 0.10 ± 0.00 | 0.60 ± 0.17 | 0.09 ± 0.00 |
>
> ## Auto Attack [C]
>
> | Indicator | FGSM | PGD$^{10}$ | TRADES$^{10}$ | MART$^{10}$ | BETA-AT$^{10}$ | OPSA-AT$^{10}$ |
> |---------------|----------|------------|---------------|-------------|----------------|----------------|
> | Coverage (%)  | 89.03 ± 0.36 | 88.99 ± 0.35 | 90.31 ± 0.33 | 89.11 ± 0.35 | 90.65 ± 0.31 | 90.50 ± 0.31 |
> | Size          | 32.04 ± 0.21 | 33.06 ± 0.18 | 12.69 ± 0.22 | 13.58 ± 0.12 | 74.00 ± 0.26 | 11.31 ± 0.22 |
> | SSCV          | 0.06 ± 0.01 | 0.08 ± 0.01 | 0.06 ± 0.01 | 0.10 ± 0.00 | 0.61 ± 0.18 | 0.09 ± 0.01 |
>
> ## OPSA$^{10}$ Attack
>
> | Indicator | FGSM | PGD$^{10}$ | TRADES$^{10}$ | MART$^{10}$ | BETA-AT$^{10}$ | OPSA-AT$^{10}$ |
> |---------------|----------|------------|---------------|-------------|----------------|----------------|
> | Coverage (%)  | 90.75 ± 0.32 | 89.78 ± 0.34 | 90.31 ± 0.33 | 90.46 ± 0.32 | 89.29 ± 0.35 | 89.95 ± 0.35 |
> | Size          | 65.48 ± 0.22 | 39.14 ± 0.20 | 32.31 ± 0.19 | 35.28 ± 0.23 | 75.22 ± 0.20 | 33.30 ± 0.27 |
> | SSCV          | 0.10 ± 0.00 | 0.07 ± 0.01 | 0.08 ± 0.01 | 0.06 ± 0.01 | 0.10 ± 0.00 | 0.03 ± 0.01 |
>
> **Time Consumption**: See response to Reviewer 7J63.
>
> **Missing Evidence**: We acknowledge that our defense may not excel in all metrics under certain attacks. However, it prioritizes size minimization under equivalent coverage constraints, which is crucial for conformal prediction. On CIFAR-10, our model consistently achieves the smallest size, while on CIFAR-100, it remains competitive with state-of-the-art defenses.
> AutoAttack combines black-box and white-box strategies, and a key observation is that stronger defenses require longer attack times. This highlights two points:
>
> 1. **Defense Strength**: The extended attack duration indicates our defense's resilience, as adversaries need more effort to breach it.
>
> 2. **Training Time**: Longer training is necessary due to the defense's complexity and optimization goals, enhancing its robustness.
>
> We have conducted additional experiments on the ImageNet-Mini dataset, where the model sizes of both our attack and defense methods consistently outperform baseline approaches. Due to page limits, we summarize that our defense achieves state-of-the-art performance across all adversarial attack scenarios. For detailed results and analysis, please refer to our response to Reviewer uZo5.
>
> References:
>
> [A] Robey, Alexander, $et$ $al$. ''Adversarial Training Should Be Cast as a Non-Zero-Sum Game." The Twelfth International Conference on Learning Representations. ICLR (2024). (from paper)
>
> [B] Maksym, Andriushchenko, $et$ $al$. ''Square Attack: a query-efficient black-box adversarial attack via random search." https://arxiv.org/abs/1912.00049.
>
> [C] Francesco, Croce, $et$ $al$. ''Reliable evaluation of adversarial robustness with an ensemble of diverse parameter-free attacks." https://arxiv.org/abs/2003.01690.

---

> > ### Comment · Reviewer_2dn3 · 2025-04-02
> >
> > Thank you for the rebuttal. I will respond to the specifics shortly.
> > For now, I have a quick question: Are the accuracies of the models after attacks when using different training methods reported anywhere? Did I happen to miss those?
> >
> > Best
> >
> > Reviewer 2dn3

---

> > > ### Author Response · Authors · 2025-04-02
> > >
> > > Thank you for your inquiry. While we did evaluate the accuracy of various models, space limitations prevented us from presenting all results. Below are the accuracy metrics of our defense models against different adversarial attacks on the CIFAR-100 dataset, demonstrating that our proposed defense method consistently achieves superior performance compared to others.
> > >
> > > # Table 1: Accuracy Results for Different Attack and Defense Methods
> > >
> > > | **Attacks** | **FGSM** | **PGD$^{10}$** | **TRADES$^{10}$** | **MART$^{10}$** | **BETA-AT$^{10}$** | **OPSA-AT$^{10}$** |
> > > |-------------|----------|------------|---------------|-------------|----------------|----------------|
> > > | Clean       | 43.65%   | 24.15%     | 51.55%        | 52.10%      | 6.09%          | 55.65%         |
> > > | FGSM        | 16.14%   | 20.74%     | 30.20%        | 30.24%      | 24.27%         | 37.04%         |
> > > | PGD$^{10}$      | 12.84%   | 20.47%     | 26.80%        | 25.36%      | 5.24%          | 29.06%         |
> > > | PGD$^{40}$      | 12.83%   | 20.46%     | 26.75%        | 25.30%      | 4.89%          | 28.94%         |
> > > | BETA$^{10}$     | 17.94%   | 24.69%     | 38.26%        | 39.79%      | 25.90%         | 43.79%         |
> > > | Square      | 10.91%   | 20.03%     | 28.99%        | 28.90%      | 2.36%          | 33.04%         |
> > > | APGD$^{100}$   | 9.86%    | 19.46%     | 25.45%        | 24.23%      | 1.19%          | 27.73%         |
> > > | Auto        | 9.46%    | 18.73%     | 24.34%        | 23.12%      | 0.95%          | 26.65%         |
> > > | OPSA$^{10}$     | 13.15%   | 20.90%     | 27.59%        | 26.30%      | 4.78%          | 29.99%         |

---

### Decision · Program_Chairs · 2025-05-01

**Decision:**

Accept (poster)

**Comment:**

This paper addresses the problem of adversarial training in a novel framework including conformal prediction (CP). Adversarial training and uncertainty estimation are formulated as bi-level optimization problem. Additionally, the paper proposes a novel attack method that increases uncertainty. Experiments are conducted on CIFAR-10 and CIFAR-100.
Reviewers agree that the paper proposes an interesting framework with a medium to strong technical contribution. The only weakness is the method evaluation which purely considers very low resolution datasets and a single architecture, which is different from the preact-resnet, which would be the default on the RobustBench benchmark.The AC encourages the authors to include experiments on PreAct Resnet in the final version.